



# On the short-term response of entrained air bubbles in the upper ocean: a case study in the North Adriatic Sea

Alvise Benetazzo[1], Trygve Halsne[2,3], Øyvind Breivik[2,3], Kjersti Opstad Strand[2],
Adrian Callaghan[4], Francesco Barbariol[1], Silvio Davison[1],
Filippo Bergamasco[5], Cristobal Molina[6], and Mauro Bastianini[1]

[1] Istituto di Scienze Marine (ISMAR), Consiglio Nazionale delle Ricerche (CNR), Venice, Italy
[2] Norwegian Meteorological Institute, Bergen, Norway
[3] Geophysical Institute, University of Bergen, Bergen, Norway
[4] Department of Civil and Environmental Engineering, Imperial College, London, United Kingdom
[5] University of Venice "Ca' Foscari", Italy
[6] Nortek AS, Norway

*Correspondence to*: Alvise Benetazzo (alvise.benetazzo@cnr.it)

**Keywords:** air bubble penetration depth; sea waves; sonar observations; air-sea gas exchange; cold-air outbreak.

**Abstract.** Air bubbles in the upper ocean are generated mainly by wave breaking at the air-sea interface. As such, after the waves break, entrained air bubbles evolve in the turbulent flow, exchange gas with the surrounding water, and may eventually rise to the surface. To shed light on the short-term response of entrained bubbles in different stormy conditions and to assess the relationships between bubble penetration depth, mechanical and thermal forcings, and air-sea transfer velocity of $CO_2$, a field experiment was conducted from an oceanographic research platform in the North Adriatic Sea. Air bubble plumes were measured using high-resolution echosounder data from an up-looking 1000-kHz sonar. The backscatter signal strength was sampled at a high resolution, 0.5 s in time and 2.5 cm along the vertical direction. Time series profiles of the bubble plume depth were established using a variable threshold procedure applied to the backscatter strength. The data show the occurrence of bubbles organised into vertical plume-like structures, drawn downwards by wave-generated turbulence and other near-surface circulations, and reaching the seabed at 17-m depth under strong forcing. We verify that bubble depths adapt rapidly to wind and wave conditions and scale approximately linearly with wind speed. A scaling with the wind/wave Reynolds number is proposed to account for the sea-state severity in the depth prediction. Results also show a strong connection between measured bubble depths and theoretical air-to-sea $CO_2$ transfer velocity parametrised with wind-only and wind/wave formulations. Further, our measurements corroborate previous results suggesting that the sinking of newly formed, cold-water masses helps bring bubbles to greater depths than those reached in stable conditions for the water column. The temperature difference between air and sea seems sufficient for describing this intensification at the leading order of magnitude. The results presented in this study are relevant for air-sea interaction studies and pave the way for progress in $CO_2$ gas exchange formulations.





## 1        Introduction

Air bubbles injected by breaking waves are ubiquitous in the upper layers of the global oceans. As such, when the wind blows over the sea above 3 m s$^{-1}$ and surface waves break entraining air, bubbles evolve in the upper-ocean turbulent flow and may eventually rise to the surface (Thorpe, 1992). This
process controls the ventilation of the ocean and the uptake of less soluble gases from the atmosphere (Deike and Melville, 2018; IPCC, 2013; Kanwisher, 1963; Woolf, 1997). For $CO_2$, it has been estimated that the ocean is a sink for ~25 % of the atmospheric gas emitted by human activities (Watson et al., 2020). During stormy conditions, a relevant part of the gas exchange is due to the bubble-mediated contribution (Reichl and Deike, 2020). The transfer of gases via bubbles depends
on the depth/time history of the bubble plume, a multi-faceted process involving advection motion in the upper sea, buoyancy of bubbles, hydrostatic pressure, and the net exchange of all gases (Woolf, 1997). For instance, in an early bubble advection model, the depth to which bubbles are carried is found to be proportional to the downward current component (Thorpe, 1982). The efficiency of the bubble-mediate transfer is controlled by the depth of the bubble penetration (Keeling, 1993), which
has been incorporated in the latest bubble gas transfer models that parametrise the kinetic term of the $CO_2$ gas exchange as a function of both wind and wave forcings (Deike, 2022). Indirectly, the importance of understanding bubble depths is also for estimating the energy dissipated by the whitecaps (Callaghan, 2018) that mediate the transfer of momentum from the atmosphere to the oceans (Cavaleri et al., 2012).

In-situ bubble plumes can be measured using acoustic instruments, which were developed for military purposes during World War II and later adapted for scientific investigations of the near-surface aerated layer (Kanwisher, 1963; Medwin, 1970, 1977). Although, at the time of writing, such type of measurements are relatively scarce, sonar observations of bubble plumes permitted to determine the local climatology of depths reached by bubbles and the relationship with surface
forcings, primarily the wind speed  (Cifuentes-Lorenzen et al., 2023; Graham et al., 2004; Strand et al., 2020; Thorpe and Stubbs, 1979; Vagle et al., 2010; Wang et al., 2016); accordingly, bubble depth has been described in the form of a linear dependency of type $\alpha(U - U_{min})$, where $U_{min}$ is the minimum speed $U$ for detecting bubbles and $\alpha$ is assumed to be a constant. Studies show that under strong winds, bubbles are easily advected to depths of several to tens of metres. It has also been proved the
importance of sea state severity and energy dissipation in determining the bubble production (Strand et al., 2020), and the dependence of bubble depth on sea states with different wave ages as a possible indicator of the breaking type (Graham et al., 2004; Wang et al., 2016). Notwithstanding, Thorpe and Stubbs (1979) suggested that the shape and depth of the bubble plumes depend on the direction of heat fluxes through the water surface, their impact has been poorly investigated in later studies
focusing only on the effect of surface wind and breaking waves on the bubble penetration.

The objective of this study is to characterise the short-term response of the bubble penetration in the upper ocean under different forcing mechanisms like wind speed, wave field, and heat fluxes. The motivation is to provide elements to improve the present understanding of the bubble plume behaviour and its connection to air-sea gas transfer parametrization and modelling. As in previous
studies, an upward-looking echo-sounding range instrument (sonar) was used to observe plumes of air bubbles, from the air-sea interface down to several meters below the surface. The sonar operated at a carrier frequency of 1000 kHz, which insonified the small fraction of bubble sizes that could be





treated as passive tracers for observing the processes in the near-surface mixed layer (Thorpe, 1992). Results are based on water and atmosphere observations collected from the *Acqua Alta* oceanographic
research platform (Figure 1). The *Acqua Alta* platform is located at 45.3° N, 12.5° E, 15 km offshore the Venice littoral (Italy) in a relatively shallow (17 m deep) and gently sloping region of the North Adriatic Sea. The area is exposed to the open sea from the southeast (*Sirocco* wind). In contrast, sea states from the northeast (*Bora* wind) are generally fetch-limited (the fetch is about 100 km). Further, mixed-wave conditions likely occur when an energetic swell from *Sirocco* leaving the central part of
the basin crosses locally-generated wind waves from the northern quadrants, mainly from the northeast (Cavaleri, 1999). Relevant to the present study is that *Bora* wind can bring cold and dry air, leading to intense sea surface heat fluxes, cooling of the water masses, and dense-water production with downward-moving waters (Benetazzo et al., 2014; Bergamasco et al., 1999; Bignami et al., 1990; Vilibic and Supic, 2006; Vilibić and Supić, 2005).

We focus on analysing average and extreme bubble penetrations and their connection with wind and wind/wave parameters in the context of the formulae used to estimate the transfer velocity of $CO_2$ gas. In section 2, we review the existing parametrizations that link bubble depth with wind speed and other forcing variables. In section 3, the sonar observations during two storms in the North Adriatic Sea (the wind speed at 10 m reached 26 m s$^{-1}$ and the significant wave height 3 m), the
method used to compute the bubble depth, and the auxiliary observations used in this study are presented. The results are presented and discussed in section 4. The main conclusions of the study are given in the final section 5. An Appendix reviewing the existing formulae used to parametrize the transfer velocity of $CO_2$ gas completes the study.

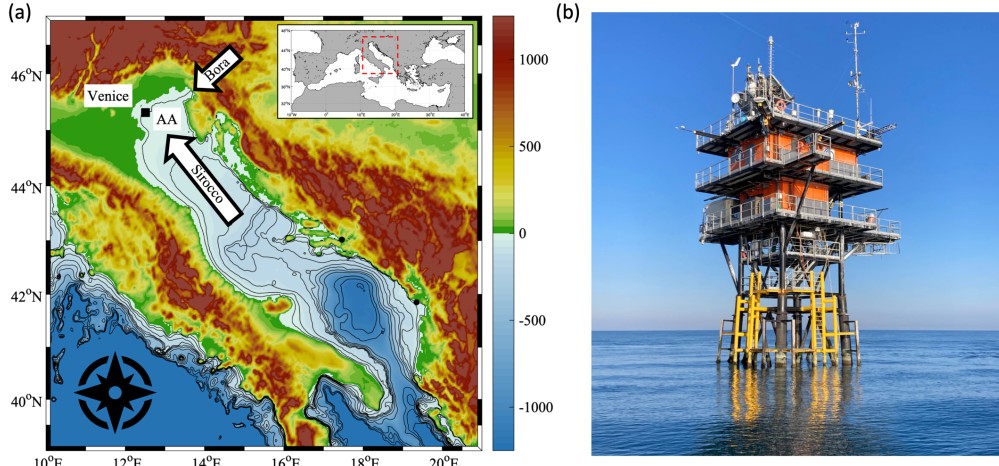

**Figure 1: The experimental site. (a) The Adriatic Sea (bathymetry in blue shading from 0 to −1250 m, in meters), surrounding orography (green-to-brown shading), and location in the north of the *Acqua Alta* oceanographic research platform (AA label). The arrows sketch the directions of the typical northeasterly *Bora* of the southeasterly *Sirocco* winds. In the inset, the Mediterranean region with at its centre the Italian peninsula. (b) Lateral view of the oceanographic research platform *Acqua Alta* (Photo credit: Francesco Barbariol).**



## 2 Parametrisation of the bubble penetration depth

In this section, we outline the parametrisations derived from experimental campaigns that provide the depth of air-bubble plumes as a function of external and internal forcings. From the early studies, it became clear that the depth of bubble plumes increases monotonically with wind speed with a high correlation (Kanwisher, 1963). This evidence suggested finding an empirical law linking bubble depth to a reference wind speed, assuming that the wind must blow over a cut-off speed for air bubbles in the bulk of water. In most wind-speed ranges, the law was found to be linear.

The earliest empirical relationship was found by Thorpe and Stubbs (1979) in the freshwater of Loch Ness Lake using records from a sonar moored at 30 m and operating at 248 kHz (the radius of resonating bubbles close to the surface is about 13 μm); these authors showed that bubbles are not detected for wind speed below $U_{min} = 2.5$ m s$^{-1}$, and that the observed average penetration depth ($z_{ba}$, in meters) of the bubble plumes increases linearly with the wind speed at 10-m height ($U_{10}$, in m s$^{-1}$), approximately as

$$z_{ba} = 0.4(U_{10} - 2.5), \qquad (1)$$

which gives, for example, $z_{ba} = 3$ m (11 m) for $U_{10} = 10$ m s$^{-1}$ (30 m s$^{-1}$). Thorpe and Stubbs (1979) recognized the relation between bubble plumes and turbulent structures within the near-surface mixing layer and that the plume variation could be associated with waves breaking in the area covered by the sonar. The authors also observed that the spatio-temporal shape of the plumes depends on the direction of the heat flux through the water surface: columnar plumes appeared when the air was colder than the water, and billow-like plumes were produced when the water was colder than the air. Hence, the bubble plumes and the thermal structure within the near-surface mixing layer are related.

The above consideration led Thorpe (1982) to include in Eq. ( 1 ) a parameter specifying the water column's stability and, hence, its ability to convey air bubbles further down when the surface water cools and water masses become denser and heavier. Since the air-minus-water temperature difference $\Delta T$ is a bulk indicator of the direction and intensity of the surface heat fluxes, the previous law was corrected for $|\Delta T| < 6°$ K as follows:

$$z_{ba} = 0.31 \Gamma_T (U_{10} - 2.5), \qquad (2)$$

where

$$\Gamma_T = (1 - 0.1\Delta T) \qquad (3)$$

is a correction factor that accounts for the stability of the water column to temperature gradients. In unstable conditions $\Delta T < 0°$ C (i.e., the air is colder than the water), the bubble plume tends to deepen further (when, for instance, $\Delta T = -5°$ C, the slope $\alpha$ of the law for bubble penetration increases from 0.31 to 0.47), while the opposite holds when the water column is stable ($\Delta T > 0°$ C). There is evidence (Thorpe, 1986a) that at wind speeds exceeding 10 m s$^{-1}$, a supra-linear relationship between $z_{ba}$ and $U_{10}$ can be more appropriate, and, at large fetches, the angular coefficient of the fitting line increases from 0.31 to 0.4. In addition, it was recognized that rain could reduce the number of breaking waves, but this effect is not currently parametrized.





Vagle et al. (2010), analysing bubble penetration data collected in the open sea (Ocean Station
Papa in the Pacific Ocean) by a 200-kHz sonar, found bubbles to depths exceeding 25 m for a wind
speed of about 22 m/s and inferred that most bubble plumes were smaller than 10 m across. The
authors confirmed the existence of a strong linear dependence between mean bubble depth and wind
speed with law:

$$z_{ba} = 0.481(U_{10} - 1.7), \qquad (4)$$

over a wide range of wind speed up to 23 m s$^{-1}$. Eq. ( 4 ) predicts $z_{ba}$ values that are 1 to 2 m larger
than those given by Eq. ( 1 ) for a given wind speed.

Wang et al. (2016) found a linear scaling for wind speeds below 10 m s$^{-1}$ by analysing 208-kHz
sonar data collected in coastal waters (depth of about 70 m, about 10 km from the coastline). Above
10 m s$^{-1}$, they verified that the relationship between wind speed and bubble depth becomes weakly
supra-linear, and found bubble depth values higher than those derived using Eq. ( 4 ) using, however,
a lower height for the reference wind speed and a bubble depth defined on the 10-min average
backscatter profile. They also noted a large variability in the bubble depth for a prescribed wind speed
since wind, it was concluded, is not the only parameter measuring the capacity of breakers to inject
bubbles in the water bulk. Following the conclusions of Thorpe (1986b, 1992), Wang et al. (2016)
argued that during the early stages of sea-state development, surface wave breaking is dominated by
plunging breakers with large bubble depths. As waves develop, spilling breakers dominate the
breaking processes with small, normalized bubble depths. These authors then included the wave effect
on bubble depths using wave age (defined as the ratio between the phase speed of the wave component
at the spectrum's peak and friction velocity in the atmospheric boundary layer). They found a clear
trend of decreasing bubble depth normalised with the significant wave height with increasing wave
age. Using wave age for scaling may be a means to improve the consistency between measurements
collected in different fetches and storm conditions (Thorpe, 1986b).

Strand et al. (2020) analyzed 70-KHz sonar data collected in northern Norway at a station a few
km offshore where the local depth is 250 m. They used a different approach to parametrize wave-
field effects on the mean bubble depth. Instead of considering the effect of bulk wave parameters,
they considered the injection of bubbles as an energetic process, which is governed by turbulent
kinetic energy flux to the sea from breaking waves $\emptyset_{ds}$; in spectral terms, this flux is given by:

$$\emptyset_{ds} = -\rho_w g \int_0^{2\pi} \int_0^{\infty} S_{ds} \, d\omega d\theta, \qquad (5)$$

where $\rho_w$ is the density of seawater, $g$ is the gravitational acceleration, and $S_{ds}$ is the dissipation
source term of the wave energy balance equation over angular frequencies ($\omega$) and directions ($\theta$). In
fully developed seas, $\emptyset_{ds}$ can be approximated as a function of the wind energy input (Craig and
Banner, 1994), proportional to the cube of the friction velocity $u_*$, while in non-equilibrium
conditions, spectral wave model result must be used for its estimate. Strand et al. (2020) found a good
correlation between mean bubble depth and $\emptyset_{ds}$ from different models (correlation coefficient
between 0.7 and 0.8); however, this was similar in magnitude to the correlation found against wind
speed and wind-sea significant wave height. On a similar argument, Cifuentes-Lorenzen et al. (2023)



found a low agreement between bubble penetration depths and wind energy going into the wave field,
       and they proposed a scaling with an effective wavelength for wave breaking.

           A different approach for characterising the depth reached by bubbles follows from the
       observation that bubbles are injected to a maximum depth comparable to the height ($H_b$) of the
       individual wave that breaks (e.g., Callaghan et al., 2013; Lenain and Melville, 2017). An upper bound

for $H_b$ is the height of the maximum waves that can be attained during stormy conditions, which is
       about $2H_s$ (Dysthe et al., 2008), with $H_s$ the significant wave height, even though most waves break
       at heights smaller than $2H_s$. In this respect, processing Strand et al. data, we have found that the
       average bubble depth is about $2.2H_s$. Maximum bubble depths were much higher and reached about
       $4H$s. Identifying bubble plumes at such deep layers can support the hypothesis that deeper plumes

are driven downward by the convergence of the Langmuir cells (Czerski et al., 2022; Plueddemann
       et al., 1996). The role of the Langmuir circulation has been, however, debated since the study by
       Thorpe (1992) and dedicated experimental campaigns are needed to judge its position against the
       direct injection induced by breaking.

## 3    Instrumentation and Methods

### 3.1  Observation of bubble plume transects using a vertical-beam sonar

The data focus of the present study were collected using a vertical-looking, high-resolution sonar
       deployed and operated in the North Adriatic Sea. The experiment spanned from 7 December 2021 to
       11 January 2022 to maximise the severity and variability of the winter storms encountered. The
       deployment includes an up-looking sonar with a typical setup for the measurement of the air bubble
       plume into the water body (Czerski et al., 2022; Gemmrich, 2010; Plueddemann et al., 1996; Saetra

et al., 2021; Strand et al., 2020; Thorpe, 1992; Thorpe and Stubbs, 1979; Vagle et al., 2010; Wang et
       al., 2016). In this study, sonar observations were made from a fixed *Signature* ADCP (Acoustic
       Doppler Current Profiler) from Nortek® operating at a monochromatic 1000 kHz pulse with a
       transmit length of 0.03 milliseconds. The instrument was bottom-mounted on a supporting framework
       that rested on the seabed at a depth of about 17 m. The sonar transducer was placed on a frame at

0.74 m above the seabed, and the blanking distance was at a vertical distance of 0.4 m ahead of the
       sensor. The backscatter signal strength was sampled at a resolution of 0.5 s (2 Hz) in time and spaced
       2.5 cm along the vertical axis. The beam width of 2.9° allows resolving surface plumes larger than
       about 0.45 m in radius. Before deployment, the instrument was prepared and calibrated by the
       manufacturer using standard procedures.



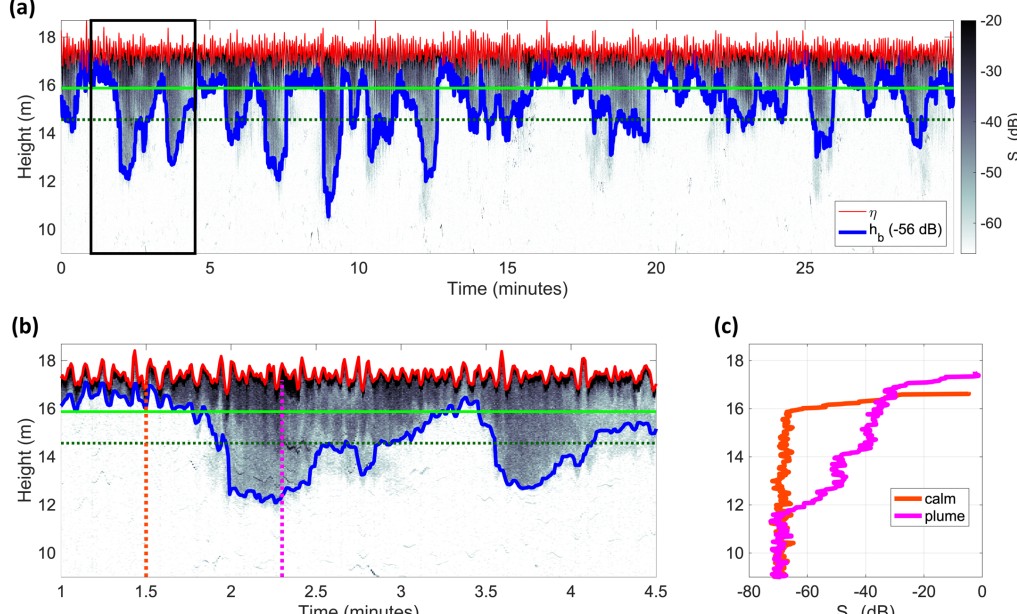

**Figure 2**: Example of measured air-bubble echo. The wind speed $U_{10}$ was 8 m s⁻¹ and the significant wave height $H_s$ = 1.3 m. (a) Time-height volumetric backscatter strength $S_v$ (grey shading, units in dB). Backscatter height (in meters) is measured upward from the sea bottom. Bubble plume height $h_b$ (blue line; threshold level of -56 dB) and surface wave elevation $\eta$ (red line). The solid light-green and dashed dark-green lines show the depth $H_s$ and 2 $H_s$, respectively. The solid black rectangle shows the echo chunk zoomed on panel b. (b) Zoom of the panel (a) between 1.0 and 4.5 minutes. (c) Vertical profiles (up to the air-sea interface) of backscatter strength during a calm period (dashed orange vertical transect in panel b) and within a bubble plume (dashed violet vertical transect in panel b).

With the echosounder mode, the sonar can measure the intensity of the echo generated after the instrument transmits a ping. The travelling time of the pulse gives an estimate of the distance in the water column to the particles reflecting the signal. With this setting, the raw echo amplitude output of the instrument is a temporal sequence of vertically distributed echo intensity (2-D time-height echograms), where the return signal is a function of the vertical distance $h$ from the instrument (752 intervals, to cover the maximum tidal range in the area) and time $t$ (3600 samples, i.e., 30-minute burst). Data bursts were acquired at the beginning of every hour (UTC), followed by 30 minutes of rest (no data). The raw echo amplitude was transformed to volumetric backscatter strength $S_v(t, h)$ using the sonar equations (Ocean Illumination, 2021).

In this study, the target sound-scattering particles are the air bubbles injected into the water bulk by wind-generated waves that break (see the example in Figure 2). After the injection phase, the population of large bubbles $O(1$ mm) rises rapidly to the surface, while other processes, such as turbulent motion, background currents, gravitational forcing, and gas exchange control the movement and density of the smallest fraction (radius < 100 µm). When the sonar-wave beam insonifies air bubbles, the incident sound wave gives rise to pulsations of the bubbles, which generate, in turn, a scattered spherical wave in the water medium. The effect is especially large when bubbles are resonant, i.e., when the eigenfrequency of their radial oscillations coincides with the sound-wave frequency (Clay and Medwin, 1977). In other words, the sound scattering in the sea is, for the most





part, due to resonant bubbles. The radius of a single ideal resonating bubble at 1000 kHz close to the surface (0-m depth) is estimated to be about 3 μm, increasing to 5 μm at 15-m depth, covering the smallest bubbles in the distribution (Randolph et al., 2014). Such bubbles rise slowly (the rise speed is below 1 mm s$^{-1}$) and may be effectively neutral in their effect on the flow and act as tracers until they dissolve. As a result, the depth of bubble plumes is mainly determined by the penetration of small bubbles, which are most susceptible to penetration due to their low-rise rate.

In the example shown in Figure 2a, bubble plumes display a growth phase, where air entrainment driven by downward forces is expected to dominate and the plume deepens to a peak depth, which is followed by a decay phase, during which the bubble rise dominates. Individual deeper plumes are not symmetric around the lowest depth, and the growth time is generally smaller than the decay time. Two distinct layers where air bubbles can evolve are visible (Czerski et al., 2022). Firstly, a shallow and near-permanent stratus layer of bubble persists from the surface to a depth qualitatively just below the significant wave height and responds naturally to the orbital motion of surface waves (see, e.g., the echo signal around 1.5 minutes in Figure 2b). This layer is sustained by wave breaking and is advected by Stokes drift and wind-driven currents.

On the other hand, at deeper depths, separate bubble plumes are located in individual cells, often in close succession and have a lifetime of roughly 30 to 120 seconds (far exceeding the period of their generation by wave-induced turbulence). The lifetime may also depend on the horizontal size of the plumes and the magnitude of the near-surface current. Thorpe (1986b) provided evidence of an increase of the bubble plumes with current and a decrease in their duration. Downward water advection may also occur at the convergent limb of Langmuir cells, which are thought to have a role in forming deep plumes (Czerski et al., 2022). Figure 2c compares the volumetric scatter strength profiles from the centre of a plume and a calm zone where no deep plume is detected. The calm zone displays scattering levels for ambient noise around -70 dB (consistent with Cifuentes-Lorenzen et al., 2023) to close to the surface (height of 16 m), from where it rapidly increases up to a maximum level. The plume profile, conversely, shows a measurable scattering level starting from about 12 m from the seabed that intensifies towards the surface. The contrast in backscattering levels between the plume and the calm zone profiles reaches 40 dB.

### 3.2 Measurement of the bubble plume depth

The volumetric backscatter signal $S_v(t, h)$ is processed to identify the temporal evolution of the bubble plume's lower edge (from the air-sea interface). This procedure is possible since the scatter strength decreases, within each bubble plume, with increasing vertical distance from the water surface, with larger $S_v$ levels in the uppermost meter. On the other hand, close to the seabed, the echo signal is dominated by the background noise. Therefore, the histogram of $S_v$ is bimodal, with a distinct separation between the two echo environments. This behaviour permits to determine bubble depth using a cut-off threshold approach, with empirical values normally ranging from -70 dB to -50 dB (Cifuentes-Lorenzen et al., 2023; Czerski et al., 2022; Gemmrich, 2010; Saetra et al., 2021; Strand et al., 2020; Thorpe, 1986b; Trevorrow, 2003; Vagle et al., 2010; Wang et al., 2016). Likewise, in this study, the depth of the bubble plume is identified with the method described below.

Firstly, the 2-D echogram $S_v(t, h)$ is smoothed using a median filter with size [3 x 3], corresponding to 1.5 seconds in time and 7.5 cm in the vertical range. Then, on the smoothed echo, the bubble height ($h_b$) is measured upward from the seabed and is defined as where the backscattered





signal falls above a threshold. The threshold is selected empirically as being not contaminated by the ambient noise (about -70 to -60 dB) and adapted for each $S_v(t, h)$ to equal min$\{\underline{S} + 10$ dB, -50 dB$\}$,

where $\underline{S}$ is the average scatter strength in the lowest third of the vertical range; this way, we account for the episodic increase of the ambient backscattering in the lowest part of the range due to fine sediments resuspension from the seabed. In practice, the selected thresholds ranged between -56 dB and -50 dB. We point out that we constrained that the signal must be continuous along the selected threshold, or else it is removed. In this manner, zones of less scatter surrounded by regions of above-

threshold scatter can be retained. An example of the identified bubble height time series $h_b(t)$ is shown in Figure 2 (solid blue line).

The thickness of the bubble layer (the bubble depth, $z_b$) is measured as the vertical distance from the instantaneous sea surface elevation $\eta$ (see next section for its determination) to the bubble height $h_b$, such that:

$$z_b(t) = \eta - h_b. \qquad (6)$$

Remapping bubble profiles to wave-following coordinates reduces the aliasing effect due to the surface wave orbital motions in determining the bubble depth (Gemmrich, 2010; Trevorrow, 2003; Wang et al., 2016). From the time series of $z_b$, two quantities are considered later in this study: (*i*) the 30-minute average ($z_{ba}$), which gauges as a whole the processes of injection, raising, and residing of bubbles in the water column; (*ii*) the 90th percentile ($z_{b90}$), which is a measure of the deeper depths

reached by the plume. The timeline of these two depths is referenced to +15 minutes of each hour, and all auxiliary variables (see next section) are interpolated over the same axis.

In evaluating vertical-looking sonar measurements of bubble plumes, it is important to recognise some elements. While it would be desirable to identify the growth and decay phases of plume evolution, the lack of information on the surface whitecap evolution and the 2-D time-height nature

of the sonar observations do not allow to distinguish between growth and decay, since the latter can be further elongated in time by successive breakings. Therefore, as in previous studies, we consider the entire time series of bubble depth to infer the relationship with environmental forcings. Moreover, the horizontal scale of bubble plumes on the water surface exceeds the width of the sonar beam, which can detect only a 1-D vertical transect of what is eminently a 3-D phenomenon. Therefore, it is not

possible to resolve the difference between bubble plumes that are locally produced and those which are advected through the vertical of the sonar beam (Czerski et al., 2022; Thorpe, 1982). In each echo record, spatial and temporal effects are not distinguished, and no information can be derived about the horizontal extent of the plume. This is the typical problem that is found in the study of 3-D ocean wave elevations from the analysis of temporal wave records (Ochi, 1998). As for waves, we assume

the ergodicity of the bubble depth process and statistical properties are evaluated from the analysis of a single record $z_b(t)$.

### 3.3 Auxiliary observations and methods

The acoustic instrument can also measure the sea elevation time series $\eta(t)$ by echo-ranging to the surface with the vertically oriented transducer (altimeter). Operationally, the surface level is defined by applying a matched filter over a series of cells to locate the maximum return signal, which marks

the well-distinct separation between water and air (red line in Figure 2, panels a and b). This operation is carried out by the instrument processor for each 30-minute burst with no intervention by the user.





The significant wave height $H_s$ is computed as four times the standard deviation of $\eta(t)$, and the wave variance frequency spectrum $S(f)$ is estimated via discrete Fourier transform using the Welch estimator with standard settings. Furthermore, a thermistor (accuracy of 0.1° C) embedded in the head
monitored the water temperature at the same temporal rate as the sonar. Originally designed to adjust the speed of sound, the near-seabed water temperature is used in this study to inspect the presence of cold-water masses. The ADCP was also used to measure the vertical components of the water velocity at 2 Hz and cell size of 0.5 m.

Sonar observations are complemented with measured atmosphere and sea data from *Acqua Alta*.
These are used to characterize the environmental forcing and bulk air-sea fluxes. The horizontal mean wind speed $U$ and direction were measured 23 m above sea level with an anemometer mounted on the *Vantage Pro2*™ weather station. The anemometer is attached to the *Acqua Alta* platform about 10 m above the structure to reduce its influence on observations. Data are averaged every 5 min (i.e., twelve values every hour), and the accuracy is 0.1 m s⁻¹ for the wind speed and 7° for the wind
direction. The same sensor provided the air temperature (accuracy of 0.5° C). The near-surface water temperature was measured 3 m below the mean sea level by a *SEA-BIRD SBE 37-SMP-ODO pumped MicroCAT*.

To identify crossing wave conditions and isolate wind waves, the directional spectrum of the wave field was measured using an additional acoustic wave and current profiler (*AWAC*) deployed at
*Acqua Alta*. Wave directions $\theta$ are based on the first pair of Fourier coefficients and are used to describe the mean direction at a given frequency. From these coefficients, the directional wave spectrum is expressed as a composition of the frequency spectrum $E(f)$ and the directional distribution $D(f, \theta)$. From this, the partitioning of the wave systems (windsea and swell) was obtained with an automated procedure of the direction/frequency spectrum based on the watershed algorithm of
Hasselmann et al. (1996), which treats the wave spectrum as an inverted catchment area, following the implementation of Hanson and Phillips (2001). The windsea partition was chosen as the one with peak direction in the same quadrant as the local wind; its significant wave height is indicated as $H_{s,ws}$.

Additional parameters that are relevant to air-sea interaction are computed using the COARE bulk algorithm (Fairall et al., 2011), version 3.6 (Fairall et al., 2022;
https://downloads.psl.noaa.gov/BLO/Air-Sea/bulkalg/cor3_6/). COARE includes the effect on fluxes of surface waves by considering in the surface drag formulation the significant wave height and phase speed of waves at the peak of the frequency spectrum. In this study, COARE is used to estimate, from measured values, the friction velocity in the air ($u_*$), the actual wind at the reference 10-m height ($U_{10}$), the neutral stability wind at 10-m height ($U_{10n}$), and the total (sensible plus latent) heat flux
across the air-sea interface ($Q$).

## 4    Results and Discussion

The results presented in this section represent the principal characteristics of the bubble depth fields measured during the experimental campaign. We aim to inspect the surface forcings influencing the bubble penetration at the scale of sea states during different storms. Further, we compare ours with
outcomes from experiments made in different sites and wind/wave conditions in search of consistency. This allows us to highlight the effect on the bubble plume scales and its deeper depths of the turbulence enhancement associated with the thermal instability of the water column. Results



are discussed in the context of the theoretical predictions of the $CO_2$ transfer velocity (total and bubble-mediated; see Appendix A) and its relationship with the penetration depth of bubbles.

### 4.1 Metocean conditions during two storms in the North Adriatic Sea

In this section, we describe the local conditions in the North Adriatic Sea for atmosphere and waves during two high-wind/wave events during which the bubble depth was measured by the sonar; specifically, the storms occurred on 8-9 December 2021 (from now on, Storm S1) and 5-7 January 2022 (from now on, Storm S2). Otherwise, during the experimental campaign, no other meaningful events were experienced (the wind speed remained below 7 m s$^{-1}$). Figure 3 sketches the model wind

speed (from ECMWF ERA5 reanalysis) over the entire Adriatic Sea at two instants during the storms, whereas the observed time series of atmosphere and wave parameters are shown in Figure 4. We anticipate that the two storms had at *Acqua Alta* similar values of total wave energy at the peak, with significant wave heights close to 3 m (in the region, storms with significant wave height at the peak of 3 m have return periods of about 1 year; Benetazzo et al., 2022). In the north, the sea-state

characteristics of the two storms were different, S1 being composed of a mixed sea (swell from SE and turning wind sea from E-NE), while S2 experienced only wind-forced waves from NE. As we shall see later, this difference leads to a different size of the bubble plume.

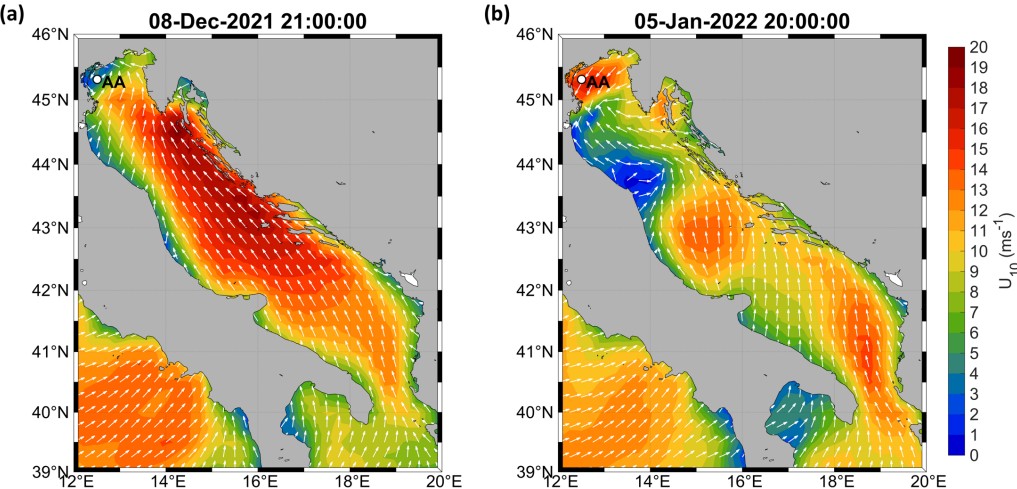

**Figure 3: Adriatic Sea 10-m height wind speed ($U_{10}$, in m s$^{-1}$) and direction (to, white arrow) during S1 (a) and S2 (b). The AA**
**label shows the position of the *Acqua Alta* oceanographic research platform. Numerical model data from ECMWF ERA5 reanalysis (https://cds.climate.copernicus.eu/cdsapp#!/dataset/reanalysis-era5-single-levels?tab=overview).**




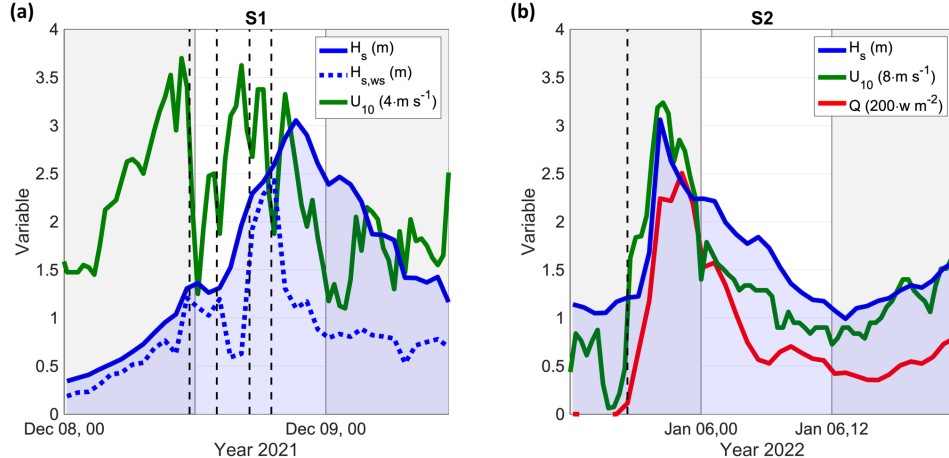

**Figure 4: Conditions of atmosphere and waves measured at the observation site. Variables: 10-m height wind speed $U_{10}$ (in meters per second), total significant wave height $H_s$ (in meters), wind-sea significant wave height $H_{s,ws}$ (in meters), and heat flux $Q$ (in W/m²). Wind speed and heat flux are linearly scaled for graphical purposes with the coefficients provided in the legend. The dashed vertical black lines show the instant when the wind turned (see main text for a detailed description). (a) Mixed-sea storm on 8-9 December 2021. (b) Unimodal, cold-air storm on 5-6 January 2022.**

As far as storm S1 is concerned (Figure 4a), the wind speed $U_{10}$ and direction (not shown) changed frequently on 8 December 2021 in the North Adriatic Sea. An alternation of northeasterly and southeasterly (from 11:30 to 14 and from 17:30 to 19 UTC) was observed (wind-direction changes in panel (a) are indicated with vertical dashed black lines). The wind speed peak of 14.8 m s⁻¹ at 11 UTC on 8 December was reached during a northeasterly phase. Following the wind, the mean wave direction was from NE (63° N, on average) until 8 December at 12 UTC; then, it turned from SE (135 °N, on average) during the remaining part of the storm controlled by a large-scale circulation forcing SE winds in mid and south Adriatic (Figure 3a). The series of significant wave height $H_s$ accompanied the growth and drop of wind and peaked at 3.1 m on 8 December at 21:30 UTC. Since then, a steady reduction of the sea state severity followed. Figure 5 shows the frequency energy spectrum $E(f)$ and the directional distribution $D(f, \theta)$ of the wave energy at the storm's peak when the sea state was a combination of wind-sea and swell. The swell from SE had a peak frequency of 0.12 Hz, whereas less energetic wind waves from NE were concentrated around 0.23 Hz. The direction of the local wind was used to isolate the wind-sea part of the significant wave height to be included in the gas transfer parametrisations.



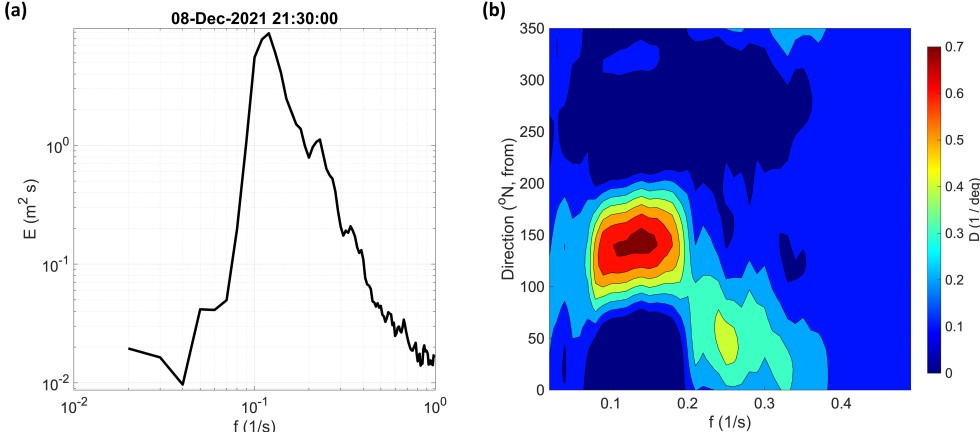

**Figure 5: Wave energy distribution measured in the North Adriatic Sea on 8 December 2022 at 21:30 UTC (storm S1). (a) Wave-frequency spectrum $E(f)$, and (b) directional distribution $D(f, \theta)$ of the wave energy.**

The storm S2 (Figure 3b and Figure 4b) was initiated by a weak southwesterly on 5 January 2022, which suddenly turned from the northeast (the incoming mean direction was 36° N), bringing cold and dry *Bora* wind in the North Adriatic Sea from 17:30 UTC. The wind speed $U_{10}$ reached 26 m s$^{-1}$ on 5 January at 20:30 UTC. The wave spectrum was unimodal, and $H_s$ peaked at 3.1 m on 5 January at 20 UTC. The atmospheric condition was the one typical of a cold-air outbreak event (Vilibić and Supić, 2005): in 10 hours, from 17 to 23 UTC on 5 January, the air temperature dropped by more than 8° C, from 11.1° C to 2.6° C. The air-water temperature difference lowered to a minimum $\Delta T$ = -7.3° C at 23 UTC. The combination of gale-force wind and large temperature difference between water and air led to surface heat flux $Q$ reaching 510 W/m$^2$ at the same time. For reference, during the 2012 record-breaking cold-air outbreak that partially iced the Venice lagoon, the heat flux at *Acqua Alta* reached 800 W/m$^2$ (Benetazzo et al., 2014). During S2, the sign of thermal vertical convection was recorded by the thermometer near the seabed. On 5 January 2022, in 30 minutes, from 20:00 to 20:30 UTC, the water temperature decreased from 11.2° C to 10.3° C. Near the seabed, the minimum recorded temperature during S2 was 10.0° C, and, after the storm, it stabilised around 10.3 to 10.5° C, about 1° C colder than before.

### 4.2 Response and scales of the bubble plume

With the method described earlier, the 30-minute time series of bubble depth $z_b$ was measured at 1-hour intervals. We focus on the two storms S1 and S2, highlighting the principal characteristics of the average ($z_{ba}$) and 90th percentile ($z_{b90}$) bubble depth. Results are shown in Figure 6.

During storm S1 (panel a), the average depth $z_{ba}$ responded at a short-time scale to the change in wind speed and direction and exceeded 3 m (3.4 m, at most) during the two phases of northeasterly wind when the wind speed was as high as 14 m s$^{-1}$. The series of $z_{b90}$ follows closely that of $z_{ba}$ (the correlation coefficient $CC$ between the two data sets is 0.99), and the extreme-to-average depth ratio $\gamma = z_{b90} / z_{ba}$ was, on average, equal to 1.8. This ratio is slightly smaller than that found by Strand et al. (2020), who used, however, the maximum value of $z_b$ as a numerator in $\gamma$. We have also found a very high correlation ($CC$ = 0.90) between $z_{ba}$ and wind speed $U_{10}$ and a very poor correlation ($CC$ =



0.15) between $z_{ba}$ and total $H_s$. The latter is due to the rapid changes in the wind direction that continuously led to non-equilibrium conditions for wave input and dissipation. Indeed, rotating winds made the energetic wave components close to the spectrum's peak largely angled from the wind, thereby receiving a small momentum. Considering only the windsea part of the wave spectrum, the correlation coefficient between $H_{s,ws}$ and $z_{ba}$ increases to 0.47, which is still smaller than values found in previous studies. During the most intense phases of the storm, the maximum $z_b$, max$\{z_b\}$, was above 6 m and peaked at 10.3 m.

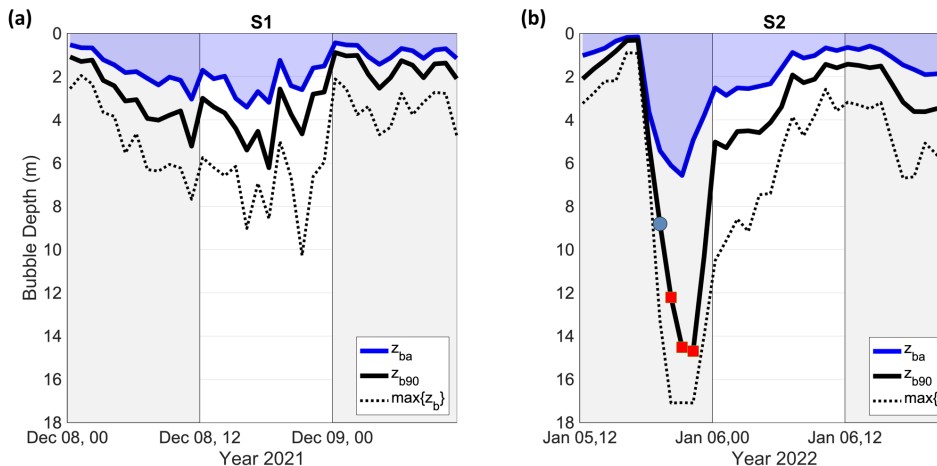

**Figure 6: Time development of the bubble depth during the storms S1 (a) and S2 (b). Values of the average ($z_{ba}$; solid blue line) and 90th percentile ($z_{b90}$; solid black line) of the bubble depth. The dotted black line shows the maximum bubble depth max$\{z_b\}$. In panel b, the red markers show instants on $z_{b90}$ when the bubble plume reached the sonar head, and the blue-silver marker shows the instant of the sonar record in Figure 7, which was taken at the onset of the cold-air outbreak.**

The vertical scales of the bubble plume were different during storm S2 (panel b), which was characterized by steady wind-sea conditions from 5 January 2022 at 17:30 UTC ahead. The average value of the bubble depth reached 6.6 m, and the correlation of $z_{ba}$ was high with both $U_{10}$ ($CC = 0.95$) and $H_s$ ($CC = 0.84$), in line with values obtained by Strand et al. (2020). Remarkably, the ratio $\gamma$ reached 3 across the peak of heat flux (from 20 to 23 UTC on 5 January), when the deepest portion of the bubble plume extended to the sonar head close to the seabed (about 17 m deep). At that time, $z_{b90}$ reached $6.1H_s$. Afterwards, on the storm's decay, the ratio $z_{b90}/z_{ba}$ diminished and was, on average, equal to 2.1, in line with values found during S1.

The backscatter of the bubble plume at the onset of the cold-air outbreak is shown in Figure 7a. It represents a period (starting on 5 January 2022 at 19 UTC, denoted by a blue-silver marker in Figure 6b) when a medium-severity sea state ($H_s = 1.7$ m) was suddenly forced by a strong and cold wind ($U_{10} = 21.5$ m s$^{-1}$, $\Delta T = -3.4°$ C) which produced individual, deep bubble plumes easily distinguishable from the background bubble population. The maximum thickness of the bubble depth reached 13 m (about $8H_s$), and $z_{b90}$ reached 8.8 m. There is a notable trend of bubble-plume deepening with time, which we estimate has a rate of about 12 m h$^{-1}$. Figure 7b shows the vertical component ($w$) of the water velocity after low-pass filtering the raw signal at 0.10 Hz to remove the wave orbital motion contribution (the peak of the wave frequency spectrum was at 0.16 Hz). This way, the vertical convection in the mixing layer is considered; we note that negative velocities (i.e., downward) around



-7 cm s$^{-1}$ accompany the creation of deeper plumes. This speed value is consistent with the prediction by Thorpe (1982), who estimated the maximum depth to which bubbles are carried as a function of the downward water current as max$\{z_b\} = 1.9w = 13.3$ m, with $z_b$ in meters and $w$ in cm s$^{-1}$. We note that stronger echo intensities are located in regions with larger vertical speeds, probably indicating locally-generated plumes.

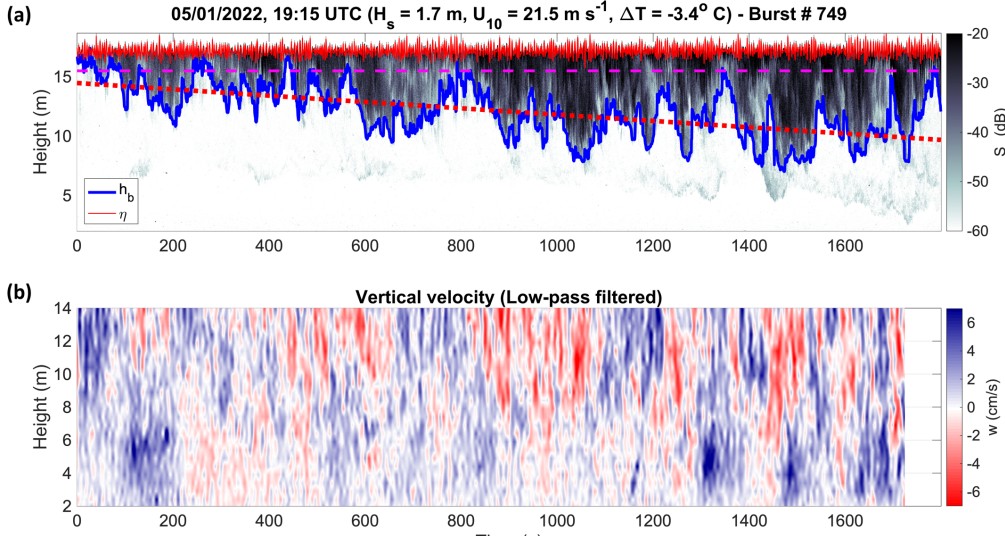

**Figure 7:** Response of the air-bubble plume at the onset of the cold-air outbreak on 5 January 2022. (a) Volumetric backscatter strength $S_v$ is in grey shading (units in dB). Backscatter height (in meters) is measured upward from the sea bottom level. The bubble plume height is shown with a blue line, and the surface wave elevation $\eta$ with a solid red line. The dashed magenta line shows the depth of $H_s$, and the dotted red line shows the line of best fit of the bubble depth. (b) The vertical component $w$ of the residual velocity (units in cm/s; positive upward) after low-pass filtering $w$ at 0.1 Hz.

Different stages of storm development led to bubble-plume shapes that can be characterised and compared to wave characteristics. In this respect, the frequency spectra of surface wave elevation and bubble height are shown in Figure 8. Data are plotted for storm S2 and correspond at instants before (panel a), at the onset of the cold-air outbreak (panel b), and after the peak of the storm (panel c). For bubble heights, two different behaviours are detectable. In the first one (panels a and c), wind waves near the peak of the wave-frequency spectrum produce breakers and displace newly produced or resident bubbles in the water body; in this case, wave and bubble spectra have similar energy levels above about half the peak frequency and, over it, spectra follow an $f^4$ law, in agreement with wave theory (Zakharov and Filonenko, 1967). Unlike waves, the bubble-height energy dominates at low frequencies with periods larger than 20 s that identify turbulent motion that effectively contributes to bubble transport. Secondly (panel b), in conditions where the vertical motion of the water bulk is also driven by thermal convection, surface-wave and air-bubble temporal scales are partially decoupled. Whereas the former preserves a typical spectral shape, the latter shows a continuous spectrum over the frequency range (no peak is evident), which decays with a milder shape proportional to $f^{-2}$.



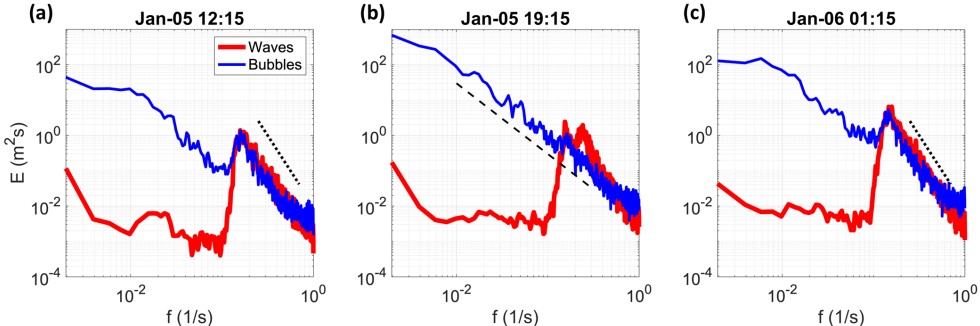

**Figure 8: Frequency spectra ($E$, in log scale) of wave elevation and bubble height at different stages of storm S2. (a) The record before the storm peak (on 5 January at 12:15 UTC), (b) at the onset of the thermal convection (on 5 January at 19:15 UTC), and (c) after the storm peak (on 6 January at 01:15 UTC). A dashed black line shows the slopes of $f^{-4}$ (panels a and c) and $f^{-2}$ (panel b).**

During storm S2, for the hours when the thermal convection was not relevant, the wave elevation $\eta$ and bubble depth $z_b$ histograms are shown in Figure 9a. Waves follow a Normal distribution closely, but data are positively skewed (the skewness coefficient is, on average, 0.14). On the other hand, bubble depths are distributed around a Lognormal distribution, as already found by Strand et al. (2020). Lognormality may suggest that the bulk of depths reached by bubbles are determined as the product of a set of independent forces at play simultaneously. Although vertical-looking sonar data do not permit clear discrimination of all factors, the temporal and spatial evolutions of entrained bubbles arise from a combination of turbulence and advection associated with breaking waves and Langmuir circulation, buoyancy of bubbles, and bubble growth or shrinking (hydrostatic pressure and net exchange of all gases). The lifetime of bubble dept $z_b$ is shown in Figure 9b during S1 and S2. The two series of lifetime show similar exponential decay with increased depth. During S2, the average $z_b$ is $1.04H_s$, close to the penetration depth considered in the bubble-mediated gas transfer model by Deike and Melville (2018). We observed a population of bubbles that get deeper than $H_s$: bubbles at depths above $2\,H_s$ have a modest lifetime (14%), and at depths above $4H_s$ are rare.

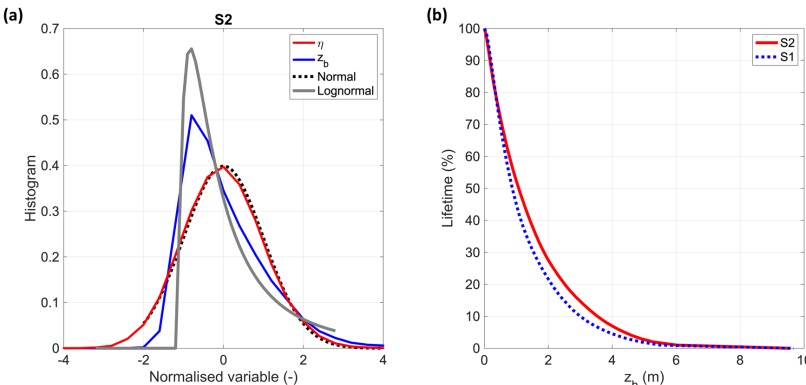

**Figure 9: Bubble depth distribution. (a) Histogram of normalised (zero mean and unitary standard deviation) wave elevations and bubble depths during S2. Normal (dashed grey) and Lognormal (solid grey) distributions are plotted for comparison. (b) The lifetime of the bubble depth $z_b$ during S1 and S2 (excluding hours of the cold-air outbreak).**





### 4.3 Bubble Depth and Surface Forcings

We consider here the observations of bubble depth and the relationship with wind speed and a combination of wind and waves. At first, bubble data are interpreted using a linear law with $U_{10}$ to compare the observed depths with predictions from parametrisations (Figure 10a). This way, the strength of breaking and any other phenomena governing the deepening and rising of bubbles are largely simplified. The average bubble depth $z_{ba}$ (in meters) is written as a function of $U_{10}$ (in m s⁻¹), and experimental data are fitted with a linear relationship in the following form:

$$z_{ba} = \alpha(U_{10} - U_{min}), \qquad (7)$$

which ensures that as the forcing term approaches a minimum threshold $U_{min}$, $z_{ba}$ approaches zero, which is what is physically expected. The dimensional scaling coefficient $\alpha$ is estimated as 0.33 and 0.31 during S1 and S2, respectively, while the onset of detectable bubbles is at a speed $U_{min}$ of 3.3 m s⁻¹ and 3.2 m s⁻¹, respectively. This minimum wind speed for having detectable bubbles is consistent with that found in previous studies on bubble and whitecap production (Hanson and Phillips, 1999; Monahan and O'Muircheartaigh, 1986). Albeit the significant variability of wind direction and speed experienced during S1, we find a good fit, being $R^2 = 0.86$. The quality of predictions improves for S2 ($R^2 = 0.91$). The law governing the average bubble depth with wind speed shows little difference between storms despite the great variability of wave conditions (mixed-sea during S1 and unimodal during S2). Moreover, no clear evidence of thermal convection effects on $z_{ba}$ during S2 exists. As a matter of fact, we have observed that the average bubble depth adapts rapidly to wind conditions and is mainly influenced by the almost continuous stratus layer of bubbles below the surface. More than $U_{10}$, surface processes are captured by the friction velocity $u_*$, whose relationship with $z_{ba}$ is shown in Figure 10b. A linear law well approximates the data scatter and the minimum friction velocity for having detectable bubbles is, on average, 5.5 cm s⁻¹.

For a given wind speed $U_{10}$, bubble depths measured here are smaller than those found in previous research, which parameterised the mean bubble depth with a similar law (Eqs. 1 to 4). For the purpose of the present study, Strand et al. (2020) measurements of $z_{ba}$ versus $U_{10}$ have been fitted with a linear law to reconcile with previous parametrisations. The law obtained is as follows (with ± 95% confidence bounds):

$$z_{ba} = 0.53 \pm 0.01(U_{10} - 1.6 \pm 0.1), \qquad (8)$$

with a coefficient of determination $R^2 = 0.72$.

Compared to the data presented in this study, only the relationship by Thorpe (1982) setting the air-water temperature difference to zero ($\Delta T = 0°$ C) tends to agree. This discrepancy suggests that existing parametrisations of bubble depth versus wind speed implicitly incorporate in the coefficient $\alpha$ the wave characteristics during the experiments used as a basis of the investigation. This indicates that the wind speed alone cannot parametrise bubble depths in all conditions, as also pointed out by Cifuentes-Lorenzen et al. (2023). In comparing data presented with others of similar characteristics, one must also note that bubble depths can depend on the sonar frequencies and echo thresholds (Czerski et al., 2022). This effect is, however, not straightforward to quantify, and we assume it is of second-order influence compared to the effect of environmental forcings generating bubble plumes.





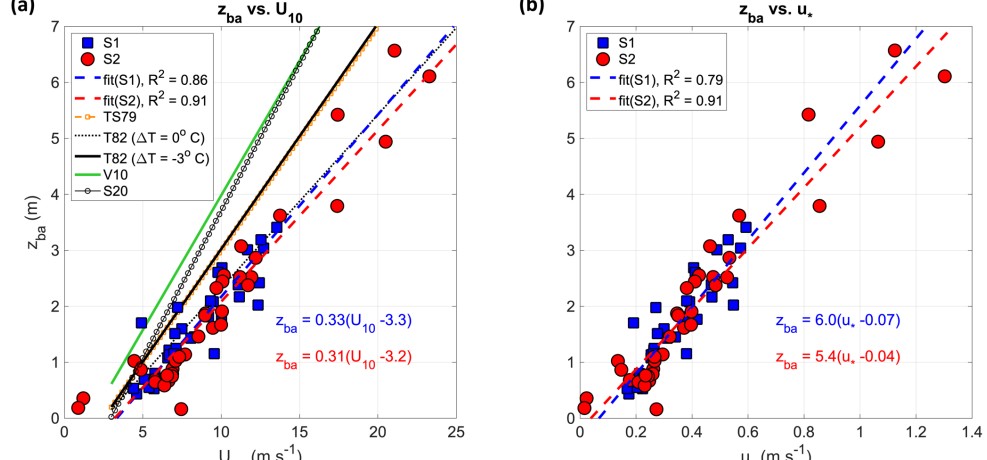

**Figure 10: Relationship between average bubble depth and wind parameters. Bubble depth $z_{ba}$ versus 10-m height wind speed $U_{10}$ (a) and friction velocity $u_*$ (b). Empirical data over storms S1 (blue marker) and S2 (red marker) and lines of best fit (dashed curve; equations in the plot area with the same colour code). Reference curves: TS79 (marked orange line), Thorpe and Stubbs (1979); T82 ($\Delta T$ = 0° C) (dotted black line), Thorpe (1982) with air-water temperature difference $\Delta T$ = 0° C; T82 ($\Delta T$ = -3° C) (solid black line), Thorpe (1982) with air-water temperature difference $\Delta T$ = -3° C (the mean temperature difference during storm S2); V10 (green line), Vagle et al. (2010); S20 (marked black line) is a fit of Strand et al. (2020) data.**

An effective way for including wave effects in the forcing process is given by equally weighting sea-state severity (say $H_s$) and wind friction velocity through the wind/wave Reynolds number $R_H$ given by (Zhao and Toba, 2001):

$$R_H = \frac{u_* H_s}{\nu_w}, \quad (9)$$

where $\nu_w$ is the kinematic viscosity of water. In a fully developed sea, the product $u_* H_s$ of water-surface processes is near-cubic with the wind speed, suggesting that wave energy dissipation plays an important role. This implies that for a given wind speed, the effect can be greater for a more developed sea (large $H_s$). Here we parameterise $z_{ba}$ in terms of $R_H$ which has been used to parameterise whitecap coverage (Brumer et al., 2017b) and which explicitly incorporates a measure of the significant wave height. . The results are shown in Figure 11. A power law fit for S2 (dashed red line) data provides overall good accuracy ($R^2$ = 0.87), with the two data having a high correlation ($CC$ = 0.92). The fit suggests that $z_{ba}$ scales closely to $\sqrt{u_* H_s}$. Extrapolating, the threshold of $f_w$ over which bubbles are detectable is 0.05 %. Formulating the sub-surface parameter $z_{ba}$ via a function of the parameter $R_H$ tends to reconcile data sets of bubble depth, as that one by Strand et al. (2020), and by Vagle et al. (2020), for which we have estimated the significant wave height using the Pierson-Moskovitz spectrum assuming fully developed seas (Pierson and Moskowitz, 1964).



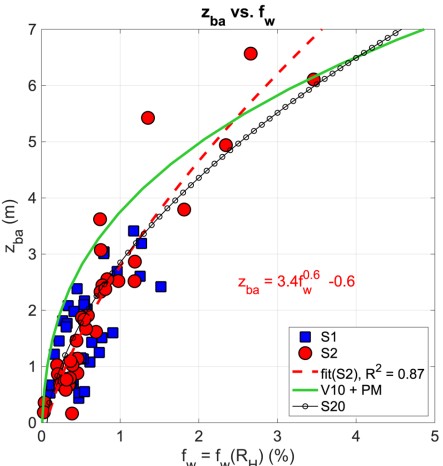


**Figure 11: Relationship between average bubble depth $z_{ba}$ and whitecap coverage $f_w$ estimated from Brumer et al. (2107b) over storms S1 (blue marker) and S2 (red marker), and the curve of best fit for S2 data (dashed red curve; the equation is given in the plot area). The solid green line depicts Vagle et al. (2010) data (V10) complemented with the significant wave height estimated from wind speed using the Pierson-Moskowitz (PM) spectrum. Strand et al. (2020) bubble data (S20) are shown with**

**a marked black line.**

The relevance of the bubble influx from the deeper plume events is described by $z_{b90}$, which is shown in Figure 12. As for $z_{ba}$, values of $z_{b90}$ closely follow a linear dependence on the wind speed for both storms, and the relationship against $U_{10}$ can be parametrised using the same law as in Eq. (7). For storm S2, we have not considered, in the fitting, values of wind speed when the heat flux $Q$

> 400 W m$^{-2}$, characterising the hours around the peak of the cold-air outbreak (colour-mapped markers and black arrow lines). For the remaining points, we find a similarity of laws derived for storms S1: $z_{b90} = 0.46(U_{10} - 2.2)$, with $R^2 = 0.77$, and storms S2: $z_{b90} = 0.46(U_{10} - 2.0)$, with $R^2 = 0.79$.

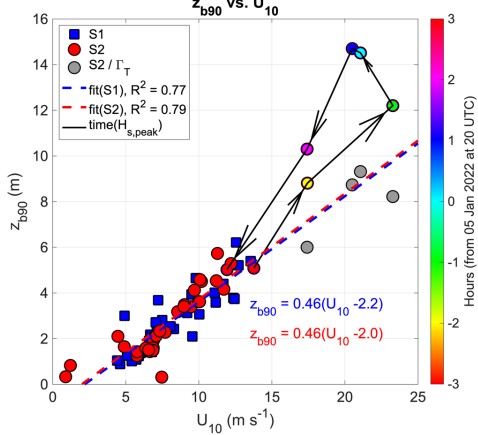

**Figure 12: Relationship between 90th percentile of the bubble depth $z_{b90}$ and 10-m height wind speed $U_{10}$. Empirical data from storms S1 (blue marker) and S2 (red marker) and lines of best fit (dashed curve; equations in the plot area with the same colour code). Colour-mapped markers and black arrow lines indicate values during S2 from -3 hours to +3 hours across the peak of heat flux $Q$. Grey markers show the same points but corrected ($z_{b90,corr}$) with the stability parameter $\Gamma_T$.**



The impact of the sinking of the water masses during the cold-air event produced extreme
depths $z_{b90}$ that were, on average, larger than those predicted in stable conditions of the water column.
Correcting measured $z_{b90}$ for the water-column stability parameter $\Gamma_T$ in the following form

$$z_{b90,corr} = z_{b90} \,/\, \Gamma_T, \qquad (\,10\,)$$

we find that the depth versus $U_{10}$ distribution agrees better with those from the other storm phases
(grey circle markers in Figure 12). Further, by following the temporal evolution of $z_{b90}$ across the
peak of $H_s$, we observe values of up to +40 % during the set-down period of the storm (2 to 3 hours
after the peak) than during its intensification (3 to 2 hours before the peak). Larger depths were
reached when the air-water temperature difference $\Delta T$ was the least. If a proportionality is anticipated
between bubble penetration depth and gas transfer velocity, this evolution across the storm peak
reverses the hysteresis cycle for the bubble-mediated $CO_2$ gas transfer velocity $k$ described by Deike
(2022). Indeed, not including the stability of the water column but only wind and wave parameters in
forcing gas transfer, Deike (2022) found values of $k$ up to a factor of two higher during the storm
intensification period than during the set-down of the same storm.

### 4.4 Bubble Depth and CO₂ Transfer Velocity

In this section, we consider the relationship between observed bubble depth and estimate of the air-
to-sea transfer velocity of CO₂ gas. The latter is here theoretically determined using forcing data
collected during the two storms. The air-to-sea exchange of CO₂ is generally calculated using a law
that relates gas flux $F$ with transfer (or piston) velocity $k$ and gas concentrations in the bulk liquid $C_w$
and at the top of the liquid boundary layer adjacent to the atmosphere $C_0$, as follows (Keeling, 1993;
Wanninkhof et al., 2009; Woolf, 1997):

$$F = k(C_w - C_0). \qquad (\,11\,)$$

By convention, $F$ is negative for a gas flux from the atmosphere to the sea. To compare variations in
diffusivity, the CO₂ transfer velocity is also given relative to the seawater temperature-dependent
Schmidt number $Sc = 660$ (the value of $Sc$ for CO₂ in seawater at 20° C) as follows (Wanninkhof,
2014):

$$k_{660} = k\left(\frac{S_c}{660}\right)^{1/2}. \qquad (\,12\,)$$

The kinematic parameter $k$ represents the gas mass transfer resistances of various physical
forcing mechanisms. It incorporates the dependence of the transfer on the diffusivity of the specific
gas in water. The transfer can be effective directly across the sea surface or between a bubble (that
encapsulates part of the atmosphere) and the water surrounding it for poorly soluble gases in rough
sea conditions. The bubble-mediated flux is most effective when bubbles reside longer and deeper in
the water volume, i.e., in stormy conditions. Since the efficiency of the bubble-mediated mechanism
depends on the pressure within bubbles, which increases with depth, a correlation is expected to exist
between gas exchange and the depth of the bubble plume.

The different processes involved in the exchange led to determining the total transfer as the
sum of the two contributions (Woolf, 1997). To describe total (diffusive plus bubble-mediated) and




relative contributions in turbulent water, two groups of parametrisations of $k$ exist in the literature (see Appendix A for a review of the formulae adopted in the present study). The first group assumes wind speed as the only kinetic forcing in the estimate, and, for the present study, we consider the

formula by Wanninkhof (2014; W14 from now on). The second group considers a combination of wind and waves, and we shall use for comparison the two formulae of the total transfer after Brumer et al. (2017a; B17, from now on), who differentiate between the total $H_s$ and the windsea ("ws") $H_{s,ws}$, and Deike and Melville (2018; DM18, from now on), and the two formulae of the bubble-mediated transfer (subscript "b") by Woolf (1997; W97 from now on) and DM18. Before going into details of

the principal outcomes, we note that coefficients in transfer velocity parametrisations should be adjusted to obtain a consistent mean value (Reichl and Deike, 2020). This calibration is not feasible in the present study because of the local and the short term of the experiment. The general behaviour, however, is preserved, and, with this caveat in mind, the results are presented below.

Figure 13 shows the time history during the two stormy events S1 (panel a) and S2 (panel b)

of total transfer velocities $k^{W14}$, $k^{B17}$, $k^{B17}_{ws}$, and $k^{DM18}$, and bubble-mediated transfer velocities $k^{DM18}_b$ and $k^{W97}_b$. As a consequence of the forcings used in the analysis, there is a remarkable difference between the two storms, such that transfer velocities from all parametrisations are about twice as larger during S2 than during S1. Moreover, differences exist among individual estimates for each storm.

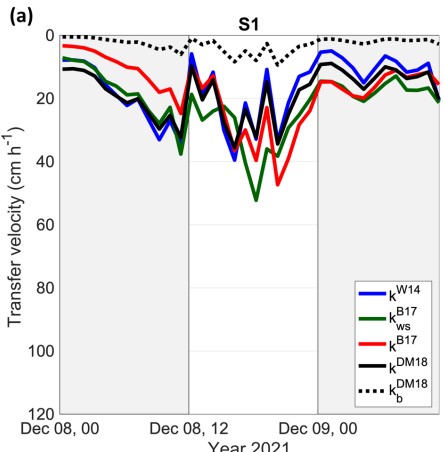
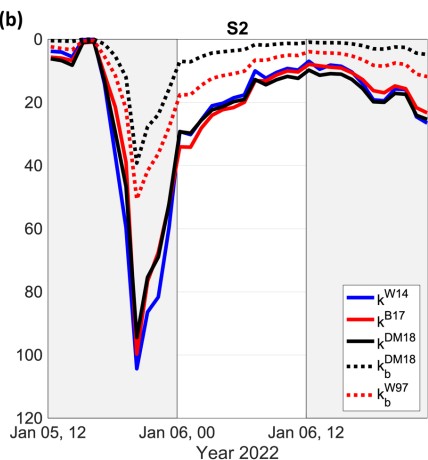


**Figure 13: Time history of gas transfer velocity of $CO_2$ during storms S1 (a) and S2 (b). Theoretical estimates of the total velocity $k$ according to parametrisations by W14 ($k^{W14}$, solid blue line), B17 ($k^{B17}$ and $k^{B17}_{ws}$, solid red and solid green lines, respectively), and DM18 ($k^{DM18}$, solid black line); Theoretical estimates of the bubble-mediated contribution to $k$: $k^{DM18}_b$ (dotted black line) and by W97 $k^{W97}_b$ (dotted red line).**

During S1, $k^{W14}$ is smaller than 40 cm/h and is consistent with $k^{DM18}$ ($CC = 0.99$ and the absolute difference is 1.5 cm/h), whereas the transfer velocity $k^{B17}_{ws}$ provide higher values that reach 50 cm/h during the most intense phase of the storm. Values of $k^{B17}$ correlate well with $k^{W14}$ and $k^{DM18}$ but display different behaviour during the growth and decay phases of the storm. The bubble-mediated contribution $k^{DM18}_b$ is below 10 cm h$^{-1}$ due to relatively small wind friction and dephasing between the wind speed peaks and wave severity. During storm S2, the three parametrisations of the






total transfer velocity $k^{W14}$, $k^{B17}$, and $k^{DM18}$ provide similar values that peak around 100 cm/h; however, $k^{DM18}$ is smaller than $k^{W14}$ (-10 cm h$^{-1}$ at the peak), conveying the fetch limitations of observed sea states, whose effect is included in DM18. The bubble-mediated term $k_b^{DM18}$ is at most 40 cm/h, slightly smaller than $k_b^{W97}$. The ratio of the bubble contribution to total gas transfer velocity, according to DM18, reaches 27 % and 42 % during S1 and S2, respectively, in line with estimates from previous studies (Deike and Melville, 2018; Reichl and Deike, 2020).

The quadratic relationship between gas transfer velocity and wind speed adopted by W14 assumes that the wind primarily induces turbulence and shear in the ocean boundary layer. Turbulence fluctuations are documented by the movement of small air bubbles, whose depths are larger when also fluctuations are larger. Based on this consideration, we investigate the relationship between bubble depths and $k_{660}^{W14}$. The result is shown in Figure 14.

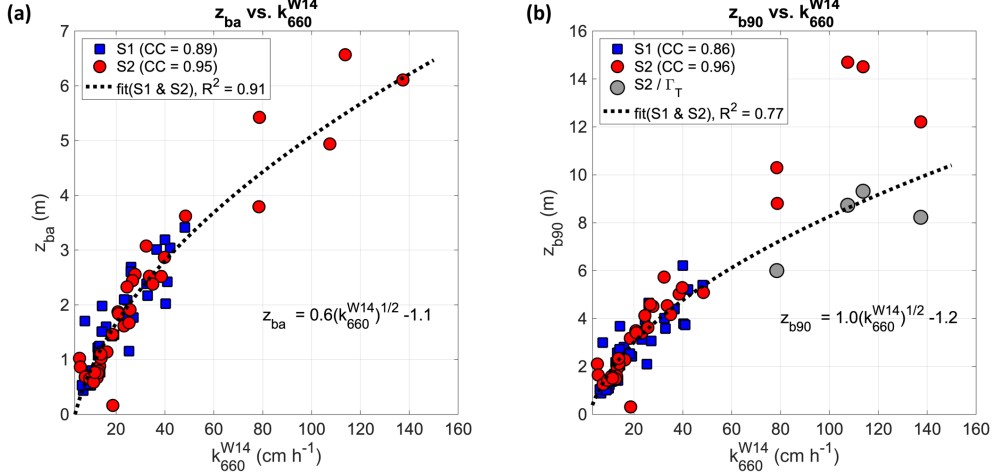

Figure 14: Relationship between wind-dependent parametrization of CO$_2$ gas transfer velocity $k_{660}^{W14}$ and average (a) and 90th percentile (b) bubble depths. Empirical data from storms S1 (blue marker) and S2 (red marker) and curve of best fit for S1 and S2 data aggregated (dotted black curve; equation in the plot area). Grey markers indicate values across the peak of heat flux $Q$ during S2 corrected with the stability parameter $\Gamma_T$.

With regard to $z_{ba}$, (panel a of Figure 14), the correlation coefficient with $k^{W14}$ is high under both storms (CC = 0.89 and 0.95 for S1 and S2, respectively), inheriting the good correspondence between $z_{ba}$ and $U_{10}$. Given that W14 assumes a quadratic relationship with $U_{10}$, and $z_{ba}$ linearly depends on $U_{10}$ with minor differences between S1 and S2, the transfer velocity $k_{660}^{w14}$ (in cm/h) is related with $z_{ba}$, (in m) with the following law:

$$z_{ba} = A\left(k_{660}^{W14}\right)^{1/2} + M, \quad (13)$$

by considering S1 and S2 data aggregated. The dimensional coefficients $A = 0.6$ and $M = 1.1$ were determined through nonlinear least-square regression (the coefficient of determination $R^2$ equals 0.91). The results reveal a rapid increase in transfer velocity with increasing bubble penetration depth. The behaviour is similar to that determined by Vagle et al. (2010), who shown the air flux $F$ associated with bubble injection versus mean bubble penetration depth. A comparison with other experiments is




not straightforward since coefficients $A$ and $M$ incorporate a sea-state dependence that is not explicit in W14. Inverting the relationship in Eq. ( 13 ), we can find an empirical relationship giving the transfer velocity as a function of the sole bubble depth, in the form of $k_{660}^{\mathrm{W14}} = f\left(z_{\mathrm{ba}}^2\right)$.

A close relationship ($R^2 > 0.85$) is also observed between $k_{660}^{\mathrm{W14}}$ and $z_{\mathrm{b90}}$ (panel b of Figure 14). As above, a fitting is made considering only S2 data for which the thermal convection has had a minor role in driving the injection of bubbles. We find that data fit well, whereas the depths across the peak of the storms are outliers. After being corrected with the water-column stability parameter $\Gamma_{\mathrm{T}}$, bubble depth values across the peak of the storm S2 tend to reconcile with others and lie closer
to the experimental curve. This result suggests a possible strategy to be used to include in the gas transfer velocity formulae the effect of the stability of the water column through the bubble depth correction, in simple term of air-water temperature difference, which can aim to larger depths reached by the bubbles and, therefore, a greater transfer of gas.

      Finally, the scatter between bubble depth and the bubble-mediated contribution to $k$ by DM18
is shown in Figure 15. Since, the bubble-mediated gas transfer increases proportionately to whitecap coverage, implying high transfer velocity in the storm conditions (Woolf, 1997), we find that a strong connection exists between $k_{\mathrm{b}}^{\mathrm{DM18}}$ and bubble depth ($CC$ up to 0.92) and that storms S1 and S2 have similar distributions, for both the average and 90th percentile of bubble depth. For both depths, a curve of the type as in Eq. ( 13 ) provides a good fit over $k_{\mathrm{b}}^{\mathrm{DM18}}$ by scaling the wind/wave forcing
term $u_*^{5/3}(gH_{\mathrm{s}})^{2/3}$ in Eq. (A.5) with a square root law. As made above for W14, the use of the parameter $\Gamma_{\mathrm{T}}$ tends to adjust deeper depths experienced by air plumes during the cold-air outbreak, albeit we find that $\Gamma_{\mathrm{T}}$ values in this case have to be reduced to about 25% to fit the experimental data.

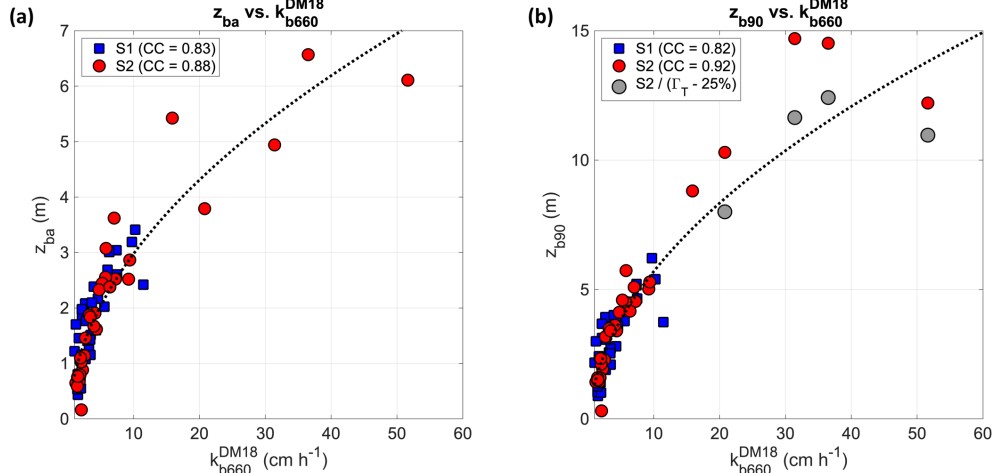

**Figure 15: Relationship between bubble-mediated parametrization of CO₂ gas transfer velocity $k_{\mathrm{b660}}^{\mathrm{DM18}}$ and average (a) and 90th**
**percentile (b) bubble depths. Empirical data from storms S1 (blue marker) and S2 (red marker) and curve of best fit for S1 and S2 data aggregated (dotted black curve; equation in the plot area). Grey markers indicate values across the peak of heat flux $Q$ during S2 corrected with the stability parameter $\Gamma_{\mathrm{T}}$ reduced by 25%**





### 5 Conclusions and Outlook

In this paper, we have investigated a data set of observations of the air-bubble plume and its penetration in the sea during two characteristic storms off the Venice littoral in the North Adriatic Sea (Italy). The analysis is made at the scale of the sea states and includes a mixed-sea (wind-wave and swell) and an unimodal wave conditions (wind-wave only). Underwater plumes have been inferred from the echo signal produced by a vertical-looking sonar operating at monochromatic 1000 kHz, which was deployed at a 17-m depth close to an oceanographic research platform that provided auxiliary observations. Bursts of 30 minutes of acoustic backscatter profiles were analyzed with the signal at 2-Hz sample rate with vertical bins of 2.5 cm. The bubble penetration was identified in the plumes with an adaptive threshold approach and was analyzed in wave-following coordinates. Two characteristic depths were investigated: the average and the 90th percentile, which have elucidated different mechanisms of the plume evolution. During the observational period, the total significant wave height peaked at about 3 m for both storms. Over the second storm, wind speed reached 26 m s$^{-1}$, and a cold-air outbreak event triggered heat fluxes up to 520 W m$^{-2}$ (the air temperature lowered to 2.6° C). This combination caused a cooling of water masses whose presence was recorded close to the sea bottom at the sonar head. In search of a prediction law for the penetration of bubbles during local storms, bubble-depth data have been parametrized against wind and wave forcings. Further, bubble plumes have been qualified in formulations of $CO_2$ air-sea fluxes using the transfer velocity as the variable of interest. The main observational findings and inference of the study are summarized as follows:

- Shallow and deep bubble plumes demonstrate a short-time response and direct connection with the surface forcings. In line with previous studies that analysed the bubble penetration depths in the ocean, we have found that bubble penetrations follow an empirical linear law with wind speed closely, although the difference in the wind-speed regime and wave development between the two storms focus of this study. The minimum wind speed for bubble plumes to be generated is around 3 m s$^{-1}$. When compared with previous parametrisations of the same type, data display smaller penetration depths for a given wind speed, which we considered as being due to the limited growth of waves in the North Adriatic Sea. When the wave forcing is incorporated in the assessment using a scaling with the wind/wave Reynolds number, a reconciliation between data sets collected at different locations and sea-state severity seems plausible.

- During the second storm (S2) focus of this study, we documented a large air-sea temperature difference that led to the cooling of waters and an intensification of the downward thermal convection. Although it is not a new phenomenon to describe, we recognise that it strongly affects the larger bubble depths, whose enhancement can be parametrised at the leading order by the air-sea temperature difference. After being corrected for the stability parameter proposed by Thorpe (1982), the bubble depth data reconciled with others measured during other phases of the same storm and those from the first storm (S1) when heat fluxes were small.

- During the intense heat-flux phase of the second storm (S2), the bubble plume reached the seabed (17 m) and its depth exceeded $6H_s$; otherwise, the maximum depths were at about half of the water column. The bubble depths followed a Lognormal distribution, suggesting that a set of independent forces are at play simultaneously that determine the depths reached by the bubbles. However, the used sonar did not permit separating the contributions from different sources.



Deeper and denser bubble plumes were accompanied by vertical convection with a downward
        maximum speed of 7 cm s$^{-1}$.

    •   The transfer velocity $k$ of $CO_2$ gas was theoretically estimated from measured data using wind-
        only and wind/wave parameterisations. During S1 and S2, values of total $k$ reached about 40 cm
        h$^{-1}$ and 100 cm/h, respectively, with small differences between predictions from the wind-
dependent Wanninkhof (2014) model and the wind/wave-dependent Deike and Melville (2018)
        model. Differences exceeding 10 cm h$^{-1}$ were found with the predictions of Brumer et al. (2017a)
        during S1. The bubble-mediated contribution was remarkable during S2, up to 40 cm h$^{-1}$,
        according to Deike and Melville (2018).

    •   By using the bubble depth as a proxy for wave breaking, we found a strong correlation between
breaking strength and theoretical $CO_2$ transfer velocity. This result was found for estimates of the
        total and the bubble-mediated contributions and to the average and extreme bubble depths, except
        for values measured during the sinking of cold-water masses. In these conditions, the scaling with
        a stability parameter provides a possible means to include the thermal convection effects in the
        total and bubble-mediated air-sea gas exchange. Results go in the direction that improved
parameterisations of gas fluxes across the air-sea surface are needed to predict the future uptake
        of these gases by the oceans.

For future research, more accurate estimates of total and bubble-mediated gas transfer will benefit
from measurements of the distribution of bubble size in the upper ocean and studies of the behaviour
of bubbles under waves. Equipment that integrates vertical-looking sonar is required, for instance,
with surface stereo cameras capable of providing breaking probability, whitecap coverage, and space-
time wave geometry. For sonars, a standardisation of the methodology to measure the edge of the
bubble plume is recommended to ease the comparison of data collected with instruments with
different carrier frequencies.

**Appendix A: Parametrisation of the air-to-sea transfer velocity of CO₂ gas**

**A.1 Wind-dependent parametrisation**

For its effectiveness and confirmed by results from laboratory and field studies with gases of low
solubility as $CO_2$, the transfer velocity $k$ in Eq. ( 11 ) has been parameterised retaining only the
influence of the momentum transfer from the wind, bypassing, in this way, the explicit role of other
processes (surface waves and turbulence, for instance). Most commonly, a quadratic wind speed
parametrisation was used (Broecker et al., 1986; Sweeney et al., 2007; Wanninkhof, 1992), which
was tuned to match the result for the global radiocarbon carbon budget over long-term timescales
(Sweeney et al., 2007; Wanninkhof, 1992, 2014). In this study, we consider the relationship with
coefficients calibrated by Wanninkhof (2014; W14) given by:

$$k^{\mathrm{W14}} = 0.251(U_{10n})^2 \left(\frac{S_c}{660}\right)^{-1/2}, \quad \text{(A.1)}$$

where the units of $k^{\mathrm{W14}}$ are in cm h$^{-1}$, and $U_{10n}$ (in m s$^{-1}$) is the neutral stability wind speed at 10-m
height. From a mechanistic standpoint, the quadratic dependencies suggest that gas exchange is





roughly related to the momentum flux at the ocean surface. The strong wind-speed dependence implies that most transfers will occur in fairly high winds, despite their relative rarity. The parametrisation W14 does not separate the contribution to the total gas flux into the bubble gas transfer and the diffusive transfer at an unbroken surface. For $CO_2$, W14 provides good estimates for intermediate winds. At low winds with a smooth water surface, the quadratic relationship will underestimate gas transfer; at high winds, bubble-mediated exchange will affect gases differently depending on their solubility, and the relationship is only suitable for $CO_2$. For their relatively simple functional form, wind-dependent formulations are used by large-scale ocean and climate communities; for instance, $k^{W14}$ is adopted for biogeochemistry numerical modelling by the European Copernicus Marine Service ([https://resources.marine.copernicus.eu/products](https://resources.marine.copernicus.eu/products)).

**A.2 Wind/wave-dependent parameterisations**

$CO_2$ flux observations display substantial scatter at moderate-to-high wind speed, and wind-dependent parameterizations tend to diverge (Brumer et al., 2017a; Deike and Melville, 2018). This effect can be attributed to the wave conditions, which can vary for a given wind speed. The complex interplay of wind and waves in determining the interaction between atmosphere and sea implies that wind speed alone cannot capture the entire variability of air-sea $CO_2$ exchange. Particularly wave breaking and its bubble production that can significantly enhance the gas exchange (Woolf, 1997). Breaking-induced bubbles offer an additional pathway for gas transfer between the atmosphere and ocean in addition to direct diffusion across the interface. Their influence increases with decreasing solubility, leading to significant enhancement of the transfer of slightly soluble gases, such as $CO_2$. The effect of breaking waves was initially considered in the bubble-mediated gas flux model by Keeling (1993). The gas transfer velocity from the bubbles to the ocean is mediated by an efficiency factor, which integrates the amount of gas each bubble transfers. The efficiency is expressed in terms of the characteristic depth of the bubble population and an equilibrium depth, and larger efficiencies, hence transfers, are obtained for greater bubble depths. In other words, deep bubble plumes provide a medium for efficient gas exchange. As for the surface manifestation of breaking-produced bubbles (whitecaps), efforts were made in the past to relate the breaking-mediated gas exchange to wind and wave forcings.

A bulk parametrization aimed at including the wave-related process in the $CO_2$ flux estimate was developed by Brumer et al. (2017a). They assumed that the whitecap fraction on the water surface is the primary process that quantifies the strength of wave breaking. The $CO_2$ gas transfer velocity data were fitted by Brumer et al. (2017a) using $H_s$ computed from the total wave spectrum as:

$$k^{B17} = 2.04 \cdot 10^{-4} R_H^{0.88} \left(\frac{S_c}{660}\right)^{-1/2}, \quad \text{(A.2)}$$

in units of m s$^{-1}$. In swell/windsea bimodal sea conditions, to account for only the active part of the wave field in generating whitecaps, Brumer et al. (2017a) isolated the windsea mode (denoted by "ws" subscript) of the wave spectrum and found the following relationship:

$$k_{ws}^{B17} = 1.64 \cdot 10^{-2} R_{H,ws}^{0.59} \left(\frac{S_c}{660}\right)^{-1/2}, \quad \text{(A.3)}$$



which suggests a near-quadratic dependence on wind speed. Brumer et al. found similar performance in determining the transfer velocity from the total or the wind-sea significant wave height and argued that non-breaking waves (from swell) contribute to the wave-induced mixing and upper ocean turbulence. Similar parametrizations that used a power law of the wind-wave Reynolds number for $k$ were developed in the studies by Zhao et al. (2003), Woolf (2005), and Jähne et al. (1985). Further, based on laboratory experiments, the dependence on the wind-wave Reynolds number was adjusted to include the wave orbital motion for scaling the efficiency of generating turbulence (Li et al., 2021).

A different approach for the characterization of the breaking-mediated flux has been considered in the spectral model by Deike and Melville (2018), in which the $CO_2$ total gas transfer velocity $k$ is separated into non-bubble ($k_{nb}$) and bubble-mediated ($k_b$) contributions, that is,

$$k^{DM18} = k_b^{DM18} + k_{nb}^{DM18}. \quad (A.4)$$

The breaking term arises from the scaling of the breaking probability density function and gives a different weight to $u_*$ and $H_s$. The most important part of the wave spectrum for gas transfer is the saturation range, where breaking dominates the energy balance. Based on this consideration, the value of $k_b$ was formulated semi-empirically by Deike and Melville (2018) using the product of bulk variables in the following form:

$$k_b^{DM18} = \frac{A_B}{K_0 R T_0} u_*^{5/3} (gH_s)^{2/3} \left(\frac{S_c}{660}\right)^{-1/2}, \quad (A.5)$$

where $A_B = 10^{-5}$ s$^2$ m$^{-2}$ is a dimensional fitting coefficient, $K_0$ is the $CO_2$ solubility in seawater, $T_0$ is the sea surface temperature, and $R$ is the ideal gas constant. The general idea behind the DM18 model is that both wind friction and waves must be present for the bubble-mediated flux to be effective. The formula in Eq. (A.5) proved efficient in reducing the scatter of wind-based gas transfer velocity parameterizations. However, other wave field characteristics related to the wave-breaking tendency, such as the wave steepness or period, were not explicitly included. The bubble contribution (A.5) to total gas transfer is stronger in high wind and wave conditions, when the wave energy and air entrainment are larger, and as the wave field develops. It is estimated that the bubble-mediated transfer is the dominant mechanism in open-ocean conditions where the wind speed exceeds 17 m s$^{-1}$ (Reichl and Deike, 2020). Recent analyses suggested modulation of the bubble-mediated contribution by current gradients, which is significant along sub-mesoscale fronts and cold filaments. There, wave breaking can be enhanced by wave-energy focus by current-induced refraction and the direct forcing by the current gradients (Shin et al., 2022).

The non-bubble contribution is led by diffusive mass transfer at the unbroken, smooth air-sea interface (i.e., no whitecapping), which wind-driven turbulence enhances. It was found to scale linearly with the friction velocity $u_*$ (Jähne et al., 1987) and is integrated in the Deike and Melville model using the COARE 3.1 parametrization (Fairall et al., 2011) as follows:

$$k_{nb}^{DM18} = A_{NB} u_* \left(\frac{S_c}{660}\right)^{-1/2}, \quad (A.6)$$



where $A_{NB} = 1.55 \cdot 10^{-4}$ is an empirical, non-dimensional coefficient (the unit of $k_{nb}^{DM18}$ is the same as $u_*$).

An alternative parameterization of the total gas transfer velocity was developed in the context of the COAREG 3.6 bulk air-sea flux algorithm (Fairall et al., 2022). In COAREG 3.6, it is assumed that the bubble-mediated transfer velocity is directly related to white capping coverage as follows (Woolf, 1997)

$$k_{b}^{W97} = B \cdot 2450 f_{w} / \left\{ \beta \left[ 1 + (14\beta S_{c}^{-0.5})^{-1/1.2} \right]^{1.2} \right\}, \quad \text{(A.7)}$$

where $B = 2.5$ is an empirical constant tuned by Fairall et al. (2022), $f_{w}$ is the whitecap fraction, and
$\beta$ is the Oswald solubility. The whitecap scaling is a surrogate for the production and mixing of bubbles. The original model by Woolf assumed the whitecap fraction scaling with neutral wind speed, while in COAREG 3.6, the wind-wave Reynolds number parameterization of whitecap coverage (%) is used in the following form (Brumer et al., 2017b)

$$f_{w} = 5.38 \cdot 10^{-6} R_{H}^{0.88} . \qquad \text{(A.8)}$$

**Data availability.** The raw data supporting the conclusions of this article will be made available by the corresponding authors upon request.

**Author contributions**. AB and TH planned the experimental campaign; AB, TH, MB, CM, and FrB
prepared the instrument deployment and participated in the data collection; AB, FrB, SD processed wave data; AB, TH, OB, KOS, FiB, AC analyzed sonar data for the bubble depth identification. All authors participated in writing the manuscript.

**Funding.** We gratefully acknowledge funding from the Italian National Research Council CNR. This activity has been partially supported by the UK Natural Environment Research Council [grant number
NE/T000309/1].

**Conflict of interest.** The authors declare that the research was conducted without any commercial or financial relationships that could be construed as a potential conflict of interest.

**Acknowledgements.** We are grateful to Dr Luigi Cavaleri (CNR-ISMAR) for valuable discussions about wind-wave effects on the Earth system and the air-sea interaction processes. Göran Brorström
from the University of Gothenburg, Sweden, is tanked for providing guidance on the processing of the echo signal. The divers of the Italian State Police helped with the deployment of the sonar at the experimental site.

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
