# Peer review of "On the short-term response of entrained air bubbles in the upper ocean: a case study in the North Adriatic Sea"

_EGUsphere, 2023_

## Author Comment (AC2)

**Revision of the manuscript**

**On the short-term response of entrained air bubbles in the upper ocean: a case study in the North Adriatic Sea**

Alvise Benetazzo, Trygve Halsne, Øyvind Breivik, Kjersti Opstad Strand,
Adrian H. Callaghan, Francesco Barbariol, Silvio Davison,
Filippo Bergamasco, Cristobal Molina, and Mauro Bastianini

Dear Editor,
Please find enclosed a detailed list of replies to the referees' comments on the paper in Subject.

The referees' comments are written in black. Our responses are written in blue colour.

We would like to express our gratitude to the two referees for the valuable comments and suggestions, which have greatly helped in improving the manuscript.

Sincerely yours,

**Alvise Benetazzo**
Institute of Marine Sciences
National Research Council
Venice, ITALY

**REFEREE #1**

The paper is well structured and provides a valuable data set with observations of bubble penetration depths. The introduction and background are appropriate and the literature review is up to date and well cited. Methods are clear (although I do request some minor changes below) and Figures are overall appropriate and support the presented narrative.

We appreciate the positive comment made by the referee on our paper. Below are our responses to the specific comments raised by the referee.

My only major issue with the paper lies in section 4.4 when the authors address gas transfer in terms of bubble penetration depths and whitecap coverage. In short, some of the main conclusions here are not fully supported by the observations and there is a perhaps too heavy of a reliance on parameterizations which are wind and wave based and addressed by the paper earlier in the previous sections. I recommend restructuring this section and presenting some of these results with a very clear caveat.

We thank the referee for providing this comment that enabled us to refine our analysis in section 4.4 and enhance the conclusions in our paper. The aim of section 4.4 analysis was to investigate the correlation between the measured depth of the bubble plume and the theoretical estimation of gas transfer velocity $k$. Unfortunately, no measure of $k$ was possible during our experiments; therefore, we relied on state-of-the-art parametrizations for $k$ (both total and bubble-mediated). The objective was not to parametrize $k$ as a function of the bubble plume depth (or vice versa) but, instead, to verify the intimate link between these two quantities in terms of surface forcings (wind and waves) that are driving mechanisms for the bubble generation and deepening and the transfer of gas. The high correlation we found between plume depth and gas transfer velocity permitted us to suggest a means for including the effect of the stability of the water column in the formulae for the gas transfer (at least for slightly soluble gases, such as $CO_2$). On the other hand, the analysis provided the basis to improve the prediction of the bubble plume penetration depth using the same forcing type and combination as gas transfer velocity. Finally, as the referee suggested (comment below), to improve the readability of the paper, we have added a Table (Table 1 of the revised manuscript) reporting the principal characteristics of the parametrization we adopted for $k$.

Specific Comments (Line by Line and Section 4.4)
Line 235-240: Why is the bubble radius increasing with depth?
For the estimate of the radius of resonant bubbles, we followed Brekhovskikh and Lysanov (1991), who provide the following formula:

Air bubbles generated by wave breaking in a relatively thin surface layer contribute considerably to sound scattering and absorption. The effect is especially large when bubbles are "resonant", i.e. when the eigenfrequency of their radial oscillations coincides with the sound-wave frequency. The resonant frequency of an air bubble in water at depth $z$ in meters is approximately given by the formula (Chap. 11)

$$f_0 = (327/a)(1 + 0.1\,z)^{1/2}\,[\text{Hz}] \qquad\qquad (1.7.1)$$

where $a$ is the bubble radius [cm].

from which we can derive, for a given frequency, how the bubble radius $a$ varies with depth $z$. As the depth increases, the radius increases as well. In the revised manuscript, we added a reference to the book by Brekhovskikh and Lysanov (1991), which was indeed missing in the initial manuscript.

Line 370: suggestion. "different size of the bubble plume" change to "different bubble plume penetration depth or extent ... " or something along the lines. This reads as the bubble size distribution is changing as a function of fetch.

We agree that this sentence may cause misinterpretation of the process under investigation. Since in section 4.1, we describe the main features of two storms in the Adriatic Sea focus of our study, we corrected the sentence referring to the short-term response of the bubble plume, in this sense considering its general temporal characteristics, which are investigated later in the paper.

Figure 4. The wind speed variable in the legend is the average wind speed over the storm?

The wind speed we measured from the oceanographic tower is an average of 5 minutes. In Figure 4, the wind speed that is shown has been interpolated over the time axis of the bubble plume observation. In the revised manuscript, we have clarified in Figure 4's caption that we are using the average wind speed.

Line 475: Please clarify. Are the authors suggesting that turbulence at the 20s period is the main driver for bubble transport?

Thanks for pointing this out. Distinct shapes of the frequency spectra $E(f)$ of the bubble penetration height are observed during the evolution of the storm S2. Low-frequency oscillations (below the frequency of the dominant wave components) were found to hold much energy (Figure 8). This behaviour is also visible in Figure 7, where the oscillations of the bubble penetration have temporal scales much larger than those of the surface waves. We argue that other than the wave motion is responsible for the large deepening of the bubble plumes, likely due to the intermittency of large breaking events and the turbulent motions in the water column. This finding aligns with the results reported by Derakhti et al. (2024). To avoid confusion, the reference to the 20-s period has been removed. Instead, low-frequency oscillations at frequencies smaller than those of surface wave motion are mentioned.

Line 465 an onward: Authors talk about bubble-height. Is this the Bubble plume penetration depth? I would suggest discussing bubble penetration depths or bubble depths.

Within the manuscript (Section 3.2), we have defined two distinct variables: the bubble height (the vertical distance of the bubble plume edge from the sea bottom) and the bubble depth (the vertical distance between the bubble plume edge and the wavy surface elevation). The two variables hold different information: the bubble height shows how the plumes are modulated by surface waves and other motions in the water column, while the bubble depth permits the analysis of how deep plumes are driven by removing the oscillations of the wave orbital motions. Within the manuscript, we have used both variables to highlight different features of the bubble plume evolution.

Line 495: typo "dept".

Many thanks for spotting this typo.

Line 495: It is not clear to me how the lifetime of the bubble depth was determined. Also, not sure how to interpret the lifetime of the bubble depth. In my opinion the temporal evolution of depth is one thing, lifetime of the bubble(s) should be addressed in terms of bubble size distributions.

The lifetime shown in the original version of the manuscript was determined by using the percentiles of the bubble plume penetration depth for both storms S1 and S2. In the revised manuscript, we have updated panel b of Figure 9 to highlight what was the message we wanted to convey, that is, how much time the bubble plume spends at a given depth and what could be the difference between storms S1 and S2. Moreover, we do agree with the referee that it would be very important to separate the lifetime of bubbles with different sizes, but this is a measure that was not available during our experiments. Therefore, we treated the lifetime as a whole, using the bubble plume depth as the only variable of interest. This point has been made clearer in the caption of Figure 9.

Regarding Figure 9a, the histogram corresponding to bubble penetration depth is for S1, S2 or both?

The histogram is for S2. In the revised manuscript, we made it clear in the caption of Figure 9a.

Line 530: Eqs. 7 and 8 I think it is relevant to state the units needed for the constant of proportionality and the physical meaning if any.

We thank the referee for pointing this out. The units in Eqs 7 and 8 are meters for the bubble depth and m/s for the wind speed; therefore, the scaling coefficient $\alpha$ has the unit of seconds (we have specified it in the revised manuscript). We have added that the coefficient indicates the effectiveness of the forcing process (wind) to displace bubble plumes under the water surface.

Line 560: define fw in the text.

We thank the referee for this suggestion; in the revised manuscript, the formula for the whitecap coverage $f_w$ (Brumer et al., 2017) function of the wind/wave Reynolds number $R_H$ has been incorporated into the main text and removed from Appendix A.

Line 550-570: Please define fully developed in wave age (or inverse wave age). Also, how do the conditions at the site compare to a fully developed wave field?

In the study of Vagle et al. (2020), no wave data were available; we have therefore assumed conditions of fully developed sea states by scaling the wave height with the wind speed. This assumption seems reasonable since Vagle et al. observations were made near Ocean Station Papa (OSP) at 50°N, 145°W, in the middle of the Pacific Ocean.

Also, I don't think I fully follow the fw argument. Are the authors exchanging the RH term in Brumer et al. 2017 from mean significant wave height to significant wave height? What is plotted in Figure 11 as fw = fw(RH)? Please clarify

The reasoning that led to the use of $f_w$ was that we wanted to scale the bubble plume depth with a parameter that incorporates both wind and waves. Moreover, since plume depth and whitecap coverage are connected through the wave-breaking phenomenon, we considered it convenient to use the Brumer et al. (2017) parametrization for the whitecap coverage using the wind/wave Reynolds $R_H$ in terms of the significant wave height (the formula is reported in the first line of Table 4 in Brumer et al). To make this point clearer, in the caption of Figure 11, we have pointed out that $f_w$ was estimated using the wind/wave Reynolds number $R_H$. The new Eq. (10) of the revised

manuscript shows how $f_\mathrm{w}$ is computed from $R_\mathrm{H}$. Moreover, Figure 11 has been updated by changing the fitting type to ensure that the bubble depth is zero when the whitecap fraction is also zero.

Regarding Brumer et al. 2017 are the authors evaluating the steepness parameter in terms of significant wave height or mean significant wave height? Same for the forcing (inverse wave age)?
Among the many formulations of whitecap coverage tested by Brumer et al. (2017), we have found it convenient to use that one in terms of the wind/wave Reynolds number expressed with the significant wave height (Eq. 9 of the manuscript); in doing this, we followed the same approach as Fairall et al. (2022).

The overall value here (in my opinion) is attempting to characterize bubble plume penetration depths as a function of forcing and wave statistics more than directly addressing the relationship between observed penetration depths and [observed] fractional whitecap coverage.
We agree with this interpretation, which reflects the content of our analysis. The use of the whitecap fraction permitted, on the one hand, the setting of a direct connection with the wave-breaking process and, on the other hand, the parametrization of the bubble plume depth through wind speed and wave variables.

Section 4.4

In my opinion it would be valuable to add a table with the gas transfer parameterizations being used (I realize they are in the appendix, but perhaps make them more available). This would also help better interpret Figure 14. This is basically a plot of observed bubble penetration depths and wind, wave forcing already presented in Fig 11. Although a very interesting point the authors attempt to make this might be pushing the dataset. The gas transfer velocity by W14 has a quadratic dependence on wind speed (as presented by the authors) Eq. 13 shows the square root of it, presenting zba as a function of wind speed. (this is identified in ~Line 665)
We agree with the referee that some overlap is present between the analysis we made on the dependence between bubble plume depth and external forcing (section 4.3) and the presentation of the results in terms of gas transfer velocity (section 4.4). However, our goal is to frame the connections between plume depth and gas transfer velocity in terms of external forcings, either wind speed only or a combination of wind speed and significant wave height. This way, the effect of the instability of the water column due to thermal gradients can be discussed in the context of gas exchange parameterizations. To avoid misinterpretation of the results, we have rephrased part of the text of Section 4.4, making it clearer the motivation of the analysis. Moreover, following the referee's suggestion, the following Table with a scheme of the different parametrizations for the transfer velocity has been added to the revised manuscript.

| | Transfer velocity | Wind and wave forcing | Reference |
|---|---|---|---|
| $k^{W14}$ | Total | $U_{10n}^2$ | (Wanninkhof, 2014) |
| $k^{B17}$ | Total | $(u_* H_s)^{0.88}$ | (Brumer et al., 2017a) |
| $k_{ws}^{B17}$ | Total | $(u_* H_{s,ws})^{0.59}$ | (Brumer et al., 2017a) |
| $k^{DM18}$ | Total | $u_*, u_*^{5/3} H_s^{2/3}$ | (Deike and Melville, 2018) |
| $k_b^{DM18}$ | Bubble-mediated | $u_*^{5/3} H_s^{2/3}$ | (Deike and Melville, 2018) |
| $k_b^{W97}$ | Bubble-mediated | $f_w$ | (Woolf, 1997) |

Table 1: Formulations adopted in this study for the estimates of the air-to-sea transfer velocity $k$ of $CO_2$ gas. Physical variables used as surface forcings: $U_{10n}$ is the neutral stability wind speed at 10-m height, $u_*$ is the friction velocity in the air, $H_s$ is the significant wave height of the sea state, $H_{s,ws}$ is the significant wave height of wind-wave partition of the sea state, $f_w$ is the whitecap fraction.

Line 665:670: "The results reveal a rapid increase in transfer velocity with increasing bubble depth penetration depth". I don't think the authors can make this claim as the data to support it is missing. We do agree with the referee, and we have then removed the sentence from the revised manuscript.

Line 680: consider restructuring this paragraph. The paragraph has been revised by improving the description of the link between bubble penetration plume depths and the transfer velocity of $CO_2$.

**REFEREE #2**

This paper describes measurements and scaling of the depth of whitecap bubble plumes from an offshore platform. The analysis links the depth to wind speed and a selection of wind/wave parameterizations of gas transfer velocity, which for weakly soluble gases are linked to bubble mediated processes.

The paper is well-written and the background material is thorough. The average depth of bubble plumes scales roughly as wind speed suggesting it is driven by the speed of the breaking waves or the wave-induced orbital velocity. The authors suggest that including wave parameters in addition to raw wind speed my improve correlations with plume depth.

In my opinion this is a good paper and it can be published essentially in its present form.

I do have a few thoughts for the authors to consider. The main issue is section 4.4 linking zb to co2 transfer velocity parameterizations. I think the link is slightly strained because k has temperature dependencies via solubility and Schmidt number that may map well to zb. Also, the use of k parameterizations that do not distinguish bubble and nonbubble modes does not make sense to me. I don't think there is any doubt that this separation is real, so I think a focus on bubble mode scaling is preferred. Just my opinion.

We thank the referee for the positive comment on our paper. As for the analysis we made in section 4.4, also following the comments from referee 1, we have revised the presentation of the results and the motivation behind the comparison of bubble plume penetration depths and gas transfer velocity $k$. As per the referee's observation, the formulae include the Schmidt number in conjunction with the mechanical forcings linked to the bubble plume depth. As for the parametrizations of the total (Wanninkhof, 2014) and bubble-mediated transfer (Deike and Melville, 2018) velocities, we have proceeded in the comparison of both terms by assuming a similar functional dependence between external surface forcings (wind and waves) and, on the one hand, gas transfer velocity and, on the other, bubble penetration depth. The high correlation we found may suggest a strategy for improving the current parameterisations used to predict bubble plume penetration depths; we also used it to infer what may be the effect of the instability of the water column (expressed in the simple term of air-water temperature difference) in the gas transfer velocity.

Here are a few editorial questions

Line 471 'bubble height' - do you mean depth?

In the manuscript (Section 3.2), we have defined and used two distinct variables: the bubble height (i.e., the vertical distance of the bubble plume edge from the sea bottom) and the bubble depth (i.e., the vertical distance between the bubble plume edge and the wavy surface elevation). The two variables hold different information: the bubble height shows how surface waves modulate the plumes, while the bubble depth permits the analysis of how deep plumes are driven by removing the oscillations of the wave orbital motions. In the analysis pointed out by the referee, we are showing the bubble height and related frequency spectra.

Line 498 What do you mean by lifetime in %? Is that the probability of a particular lifetime? Also, Fig. 9b is referred to as 'lifetime'. do you mean probability distribution of lifetime?

Following the referees' comments, we have improved the description of the lifetime and its representation in the new Figure 9b. In the revised version, the lifetime in hours is provided to avoid confusion on how the lifetime % was calculated.